# Regularities in species' niches reveal the world's climate regions

**Joaquín Calatayud[1,2]\*, Magnus Neuman[1], Alexis Rojas[1], Anton Eriksson[1], Martin Rosvall[1]**

[1]Integrated Science Lab, Department of Physics, Umeå University, Umeå, Sweden; [2]Departamento de Biología, Geología, Física y Química inorgánica, Universidad Rey Juan Carlos, Madrid, Spain

**Abstract** Climate regions form the basis of many ecological, evolutionary, and conservation studies. However, our understanding of climate regions is limited to how they shape vegetation: they do not account for the distribution of animals. Here, we develop a network-based framework to identify important climates worldwide based on regularities in realized niches of about 26,000 tetrapods. We show that high-energy climates, including deserts, tropical savannas, and steppes, are consistent across animal- and plant-derived classifications, indicating similar underlying climatic determinants. Conversely, temperate climates differ across all groups, suggesting that these climates allow for idiosyncratic adaptations. Finally, we show how the integration of niche classifications with geographical information enables the detection of climatic transition zones and the signal of geographic and historical processes. Our results identify the climates shaping the distribution of tetrapods and call for caution when using general climate classifications to study the ecology, evolution, or conservation of specific taxa.

## Introduction

Climate governs the basis of life on Earth. Besides historical contingencies and geographical barriers, abiotic conditions determine species ranges (*Woodward, 1987*; *Hoffmann and Parsons, 1997*; *Calatayud et al., 2019b*) and derived diversity patterns (*Hawkins et al., 2003*; *Kreft and Jetz, 2007*; *Mendoza and Araújo, 2019*). On a global scale, distinctive climate types impose generalized constraints that shape species pools adapted to particular climates (*Whittaker, 1962*). Identifying the boundaries of these climate types is a fundamental challenge to understanding how life organizes on Earth, with acute implications in diverse disciplines given the current climate change.

Pythagoras proposed a classification of climate regimes of the known world in the 6th century BC (*Sanderson, 1999*). Still, it was not until the 19th century that geographers laid the foundations for such classifications (*Oliver, 1991*). By that time, researchers noticed the close relationship between the distribution of various life forms, especially vegetation types, and climate (*Oliver, 1991*). For instance, Köppen built his long-standing climate classification from pioneer plant classifications, assuming that vegetation types carry information about climatic conditions (*Kottek et al., 2006*; *Thornthwaite, 1943*). This assumption has received considerable support (*Rohli et al., 2015*), and researchers and stakeholders use Köppen's classification system in a wide range of disciplines, including climatology (*Spinoni et al., 2015*), geography (*Gentine et al., 2012*), conservation planning (*Tobin et al., 2014*), and ecology (*Garcia et al., 2014*).

But the general climatic conditions derived from plant species may not apply to other organisms. Plants' and animals' climatic determinants will likely differ, given their large physiological differences. Although diversity patterns of plants and animals are relatively congruent, the climatic correlates of these patterns vary (*Whittaker et al., 2007*; *Qian and Ricklefs, 2008*). Hence, if climatic adaptations vary among taxa, so will the boundaries defining their climate types. In *Thornthwaite, 1943* words,

\*For correspondence:
j.calatayud.ortega@gmail.com

**Competing interests:** The authors declare that no competing interests exist.

**eLife digest** There are many distinct climates on Earth, from tropical savannas and temperate forests to dry deserts. Historically, each region has been defined by how its annual weather patterns shape the type of vegetation present. For example, hot and humid environments support the growth of evergreen forests that would not survive in drier places.

Identifying the boundaries between climate regions is key to understanding how life is organized on Earth and predicting how climate change will affect different species. Current climate classifications, however, do not account for where animals can be found or how local conditions, such as precipitation and average temperatures, shape the distribution of different animal species. To bridge this gap, Calatayud et al. analyzed the preferred climate of about 26,000 animal species, including amphibians, birds, mammals and reptiles.

For each species, Calatayud et al. calculated the annual rainfall and temperature of its local environment, or 'niche', using previously collected data. They then used a computer algorithm to group together climates that had similar species. This identified 16 climate regions which govern the distribution of the animals studied. Calatayud et al. found that these newly defined climatic regions resembled some of the regions classified using plants. This was particularly true for high-energy climates that had lower levels of rainfall and hot temperatures, such as deserts and the tropical savanna.

The animals and plant species living in high-energy regions were found to be fairly consistent across both classification systems. Whereas the species present in milder and colder climates, such as temperate forests or Mediterranean climates, were found to be much more varied. This suggests that temperate climates are harder to classify and may affect the distribution of plants and animals differently. It also implies that less extreme conditions support a larger range of species than harsher climates in which only species with certain adaptations are able to survive.

These findings build the basis for a better understanding of how climates shape ecosystems. More specific climate classifications, based on such analyses, could be used to inform conservation strategies for animal species in the face of climate change.

the 'truly active factors' describing a climate type may vary among organisms, and it remains unknown whether Köppen's climate classification can indicate the active climatic factors for other organisms than plants. Despite several attempts to refine or propose alternative climate types or regions (*Trewartha, 1954*; *Holdridge, 1947*; *Thornthwaite, 1948*; *Netzel and Stepinski, 2016*; *Gardner et al., 2020*), quantitative studies defining climatic regions for other organisms are still lacking.

The current information on species distributions and global climatic variables, together with recent advances in niche modeling and classification techniques, provides an unprecedented opportunity to identify the climatic boundaries shaping the distribution of faunas and floras across the globe. The last decades have witnessed a tremendous collective effort to record occurrences of a large number of species (*GBIF, 2019*), which has resulted in comprehensive datasets of the distributional ranges of several groups of species (*IUCN, 2015*; *BirdLife, 2015*; *Roll et al., 2017*). Also, data on climatic variables at a global scale have been developed at high spatial resolutions (*Fick and Hijmans, 2017*; *Trabucco and Zomer, 2009*). Combining these two sources of information, we can characterize the realized climatic niches of different taxa and find regularities among them. For example, projecting these realized climatic niches into a climatic space (*Broennimann et al., 2012*) should, if climatic boundaries exist, reveal co-occurring groups of species across particular portions of the climatic space. Thus, identifying these niche domains should uncover the main climatic boundaries shaping the organization of life (*Figure 1*).

In addition to the climate, dispersal barriers and historical contingencies influence the shape of niche domains (*Warren et al., 2014*; *Calatayud et al., 2016*; *Calatayud et al., 2019b*). Therefore, similar climates can have different effects across different geographic regions (*Ricklefs, 1987*). For instance, while a given climate in some parts of the Earth may lead to specific species pools, the same climate in other parts of the Earth may not. Such a potential lack of specific species can occur, for example, because the required adaptations have not appeared (*Flohr et al., 2013*), the adapted

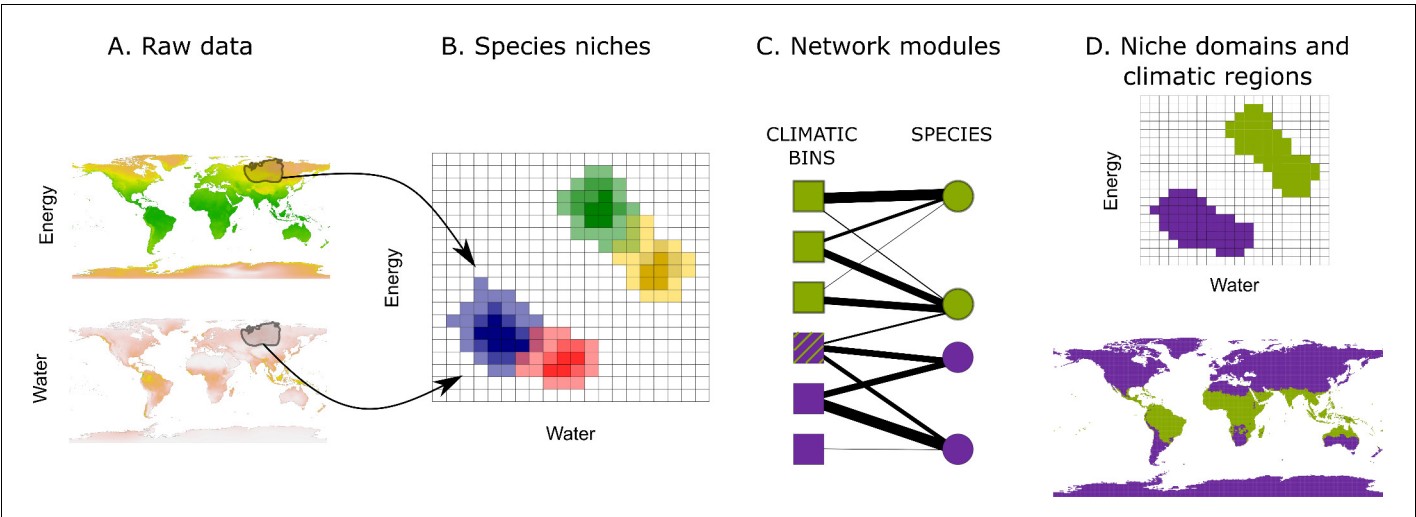

**Figure 1.** Workflow to identify niche domains and climatic regions. Using the climatic conditions a given species experiences within its range (**A**), we project species niches into a climatic space discretized in an optimal number of bins (Appendix 1) (**B**). We translate the binned data into a weighted bipartite network, where climatic bins and species form the nodes and the probabilities of finding the species in the bins form the weighted links (**C**). Using a network community detection algorithm, we identify domains of the climatic space with similar species (**D**, upper). The climatic conditions defining these domains delineate the corresponding climatic regions of the Earth (**D**, lower). The striped climatic bin links to species classified in both climatic domains, and, therefore, it represents a diffuse transition with low specificity.

species have not been able to disperse (*Tuomisto et al., 2003*), or the area is too small to hold large species pools (*Connor and McCoy, 1979*). Studying the signature of these historical and geographical processes, known as geographical signals, in niche domains can provide valuable information about the potential mechanisms behind them and their associated climatic regions.

We explore the global climate regions of tetrapods by characterizing the climatic niche domains of amphibians, birds, mammals, and reptiles. Tetrapods are a well-suited group for our purpose. First, comprehensive databases are available, including the distributional ranges of most species in the group (*IUCN, 2015*; *BirdLife, 2015*; *Roll et al., 2017*). Second, the different classes of tetrapods have diverse capabilities to disperse and withstand abiotic conditions, allowing us to investigate whether various capabilities influence climatic classifications. Third, accumulated evidence about the main climatic factors controlling the distribution of these species simplifies the selection of appropriate climatic variables. In particular, the distribution of tetrapods is strongly determined by the water and energy aspects of climate (*Hawkins et al., 2003*; *Currie, 1991*; *Tingley et al., 2009*; *Gouveia et al., 2014*; *Pie et al., 2017*; *Cooper et al., 2011*). Finally, researchers study tetrapods in several disparate fields – from animal husbandry (*Abecia et al., 2017*) to ecological (*Englert Duursma et al., 2019*) and evolutionary studies (*Rolland et al., 2014*) – where a description of their climatic regions can be especially useful.

In our classification approach, we project the realized niche of each tetrapod species onto a binned two-dimensional space representing water and energy, the primary climatic factors impacting their geographic distribution (*Figure 1*). After translating this climatic niche space into a weighted bipartite network, we apply a network community detection algorithm to identify climatic niche domains with similar species pools. Finally, by mapping the climatic niche domains back onto the Earth's surface, we provide the climatic regions.

The detected climatic regions support the notion that similar climatic determinants underlie animal and plant distributions in high-energy regions, including deserts, tropical savannas, and steppe regions. However, differences in temperate climates across all groups indicate that specific climatic regions for each group of taxa are required to address ecological, evolutionary, and conservation questions.

## Results

### Classification of climatic niches

We first identified the species niche domains by calculating the proportion of observations of each species within each bin of a two-dimensional climatic space defined by potential evapotranspiration (PET) and annual precipitation (AP; *Figure 1*, 'Materials and methods' and Appendix 1). We represented this data as a weighted bipartite network, where climatic bins and species form two disjunct sets of nodes and the probabilities of finding the species in the bins form the link weights. Using a hierarchical network clustering algorithm (*Rosvall and Bergstrom, 2008*; *Rosvall and Bergstrom, 2011*), we obtained groups of climatic bins holding similar species, niche domains, and the species associated with them.

We found similarities among tetrapods classes in the detected niche domains but also observed some differences (*Figure 2*). For instance, the number of major domains with 50 or more species in the lower hierarchical level is similar (ranging from 13 to 15) across tetrapods classes. However, mammals and birds show a domain of low-energy inputs, whereas reptiles present some domains across arid conditions, that is, with high energy and low water availability (*Figure 2*). Finally, we classified the climatic space of tetrapods by using all species jointly. The clustering algorithm divided the niche space of tetrapods into 16 main domains that are similar to those of the independent classes, and some of the particularities described above did not appear (*Figure 2*).

Since uncertainties related to the ranges of species exist, we employed a bootstrap and a significance clustering procedure (*Rosvall and Bergstrom, 2010*; *Calatayud et al., 2019a*) to assess the domains' robustness ('Materials and methods'). While several domains were well supported, we

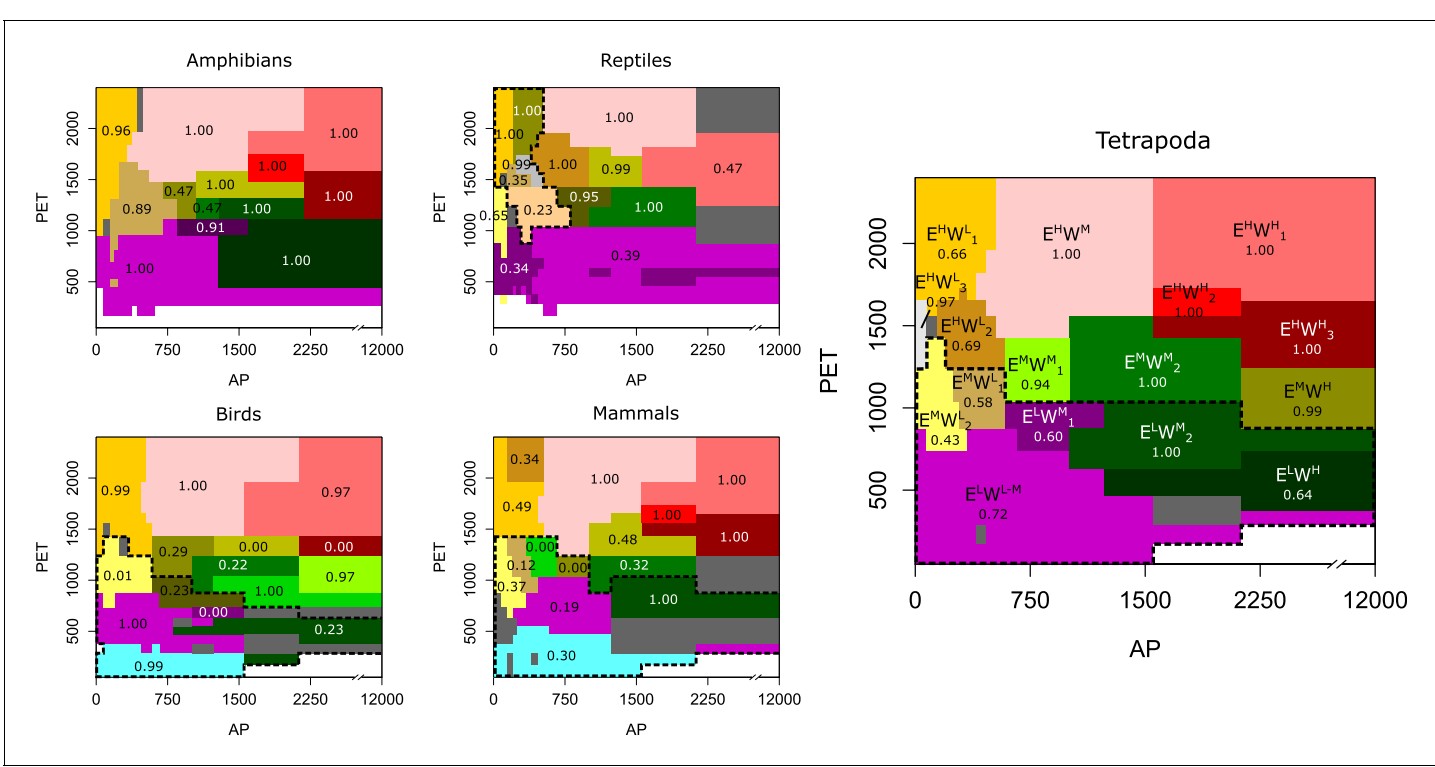

**Figure 2.** Tetrapods' niche domains across the climatic space. The climatic niche domains of each group shown across a space defined by potential evapotranspiration (PET) as a surrogate of energy and annual precipitation (AP) as a surrogate of water inputs. Tetrapods' domains are labeled so that E and W represent energy and water, respectively, and superscripts H, M, and L mean high, medium, and low, respectively. Numerical subscripts differentiate domains of similar climates. Numbers between 0 and 1 indicate the bootstrap support. The dotted line represents the domains at the highest hierarchical level. Domains of less than 50 species are colored dark gray. Domain colors across groups only indicate similar climatic regions. To characterize the climatic space, we used 17 divisions of both PET and AP for all groups except for amphibians, where we used 18 (see Appendix 1). The online version of this article includes the following source data for figure 2:

**Source data 1.** niches domains and the species associated to them.

found that the niche domains corresponding to intermediate energy (between approximately 1000 and 1500 PET units; $E^M$ climates in *Figure 2*) and low to moderate water (up to approximately 800 ml; $W^L$ to $W^M$) were less robust. This robustness analysis shows that these niche domains are more challenging to classify.

## Animal vs previous plant-based climate regions

While Köppen's climate classification is based on expert knowledge on vegetation physiognomy and the distribution of vegetation types, its wide use makes it worthwhile to compare the climate regions derived from this classification system to the ones produced here. Hence, we studied the geographic location of the climatic conditions associated to niche domains, the climatic regions shown in *Figures 1* and *3*, which allowed for a more precise comparison between groups and Köppen's plant-based regions. The similarities among the regions of tetrapods classes measured as adjusted mutual information (AMI) ranged from 0.57 to 0.68, with mean AMI = 0.62 (*Table 1*). Moreover, the regions based on the niche domains of all tetrapods together were to some extent congruent with the regions of its independent classes (mean AMI = 0.71, ranging from 0.66 to 0.77). Köppen's regions were more dissimilar to the regions of all tetrapods together (AMI = 0.44) and the regions of each class of tetrapods independently (mean AMI = 0.44, ranging from 0.40 to 0.47).

Focusing on particular regions, we saw that climates of high energy ($E^H$) were consistent among tetrapod groups and Köppen's classification. Desert climates (high energy and low water, $E^H W^L$, *Figure 3*, and *Appendix 1—figure 1*) were the most similar across all groups. Tropical savanna and steppe climates (high energy and medium water, $E^H W^M$) were also consistently defined, though both of these Köppen regions were classified together for all groups but reptiles (Figures 2 and 3 and *Appendix 1—figure 1*). Similarly, Köppen's tropical rainforest and tropical monsoon climates were for the most part well recovered (*Appendix 1—figure 1*). However, we found three different tropical-humid regions, each one mostly corresponding to one of the three larger masses of tropical rainforests: Amazonian, African, and Southeast Asian rainforests; $E^H W^H_1$, $E^H W^H_2$, and $E^H W^H_3$, respectively (Figures 2 and 3). Regarding regions of low energy, we found a slightly higher level of disagreement between Köppen's and tetrapods' regions (*Figure 3*). Finally, temperate climates (medium energy $E^M$) were the least congruent between tetrapod groups and Köppen's regions. These regions of medium energy were at the same time the least congruent among groups and the least supported by the bootstrap analyses, suggesting that these climates could impose less restrictive conditions in general and allow the appearance of idiosyncratic and variable adaptations.

## Climatic transition zones

A complete understanding of niche domains and their associated climatic regions entails the exploration of whether their boundaries represent abrupt or diffuse transitions. Climatic conditions corresponding to diffuse transitions should present low specificity levels to the domain where they belong (*Figure 1*). Hence, we can consider specificity as the opposite of transitivity. Our network approach allows us to calculate this specificity by the dual classification of climatic bins and species into the same niche domains (*Figure 1*). We computed the specificity of each climatic bin as the ratio between the link weights of the species classified in the same domain and the total link weights (*Bernardo-Madrid et al., 2019*; *Calatayud et al., 2019b*). Then, we projected these values geographically. As expected, lower specificity values were in general associated with the boundaries of the climatic regions (*Figure 4A* and *Appendix 1—figure 2*). Beyond boundaries, our results also revealed that harsh conditions, such as desert and continental-polar climates ($E^H W^L$ and $E^L W^L$), present the highest specificity levels, regardless of the group (*Figure 4A* and *Appendix 1—figure 2*), reflecting the difficulty of colonizing these climates. Conversely, temperate regions showed the lowest levels of specificity. These regions were also weakly supported in the bootstrap analyses, and we found that bootstrap p-values and mean specificity were significantly correlated (standard generalized linear mixed model [GLMM] coeff. 6.21; $p < 0.001$ conditional = 0.29, see 'Materials and methods'). Together with the higher variability of these regions across groups, this result further supports the idea that these climatic conditions could impose less restrictive conditions to tetrapods.

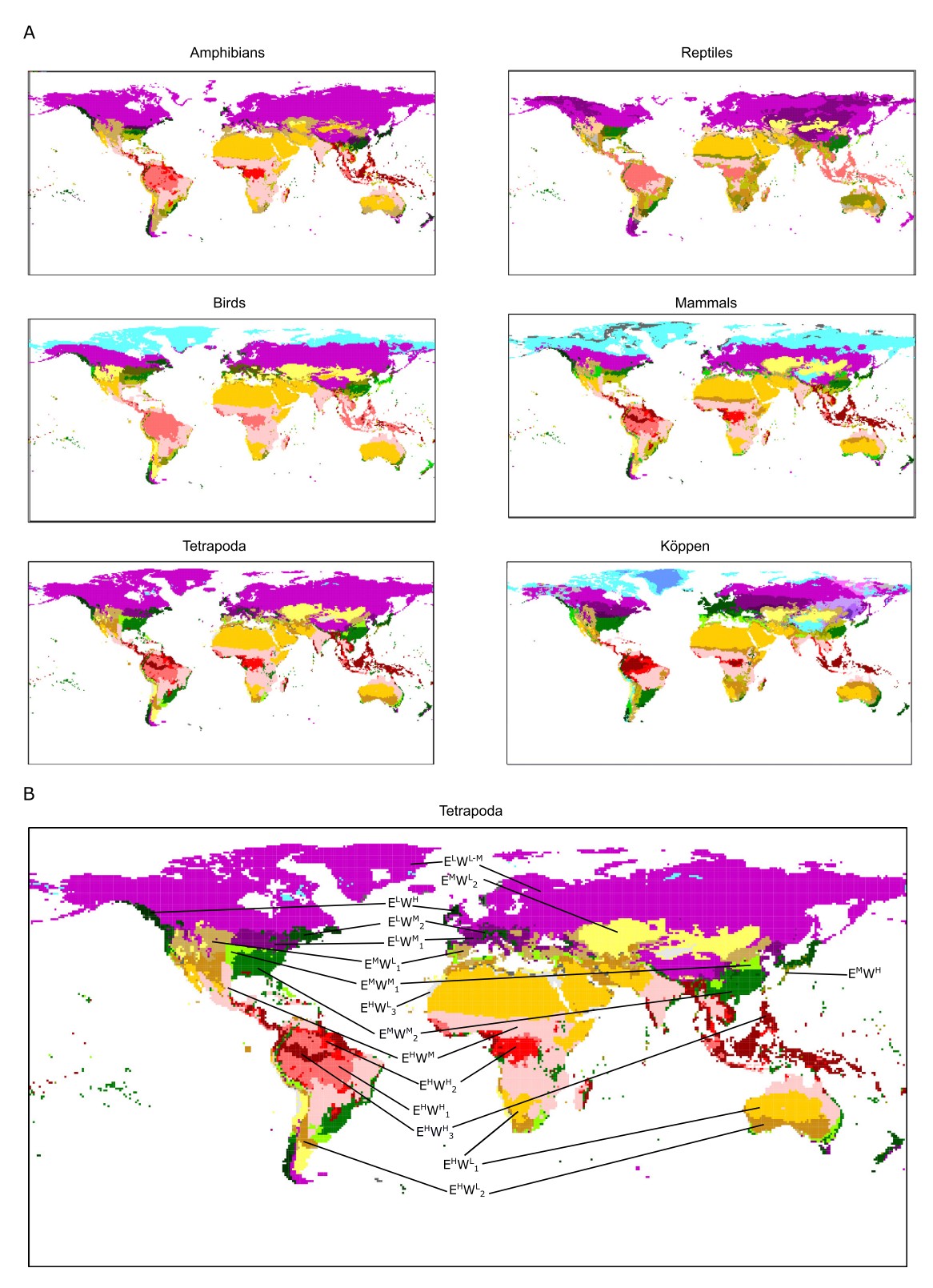

**Figure 3.** Tetrapod groups and Köppen's climatic regions. (**A**) Geographic location of tetrapods' niche domains and Köppen's climatic regions. See also **Appendix 1—figure 1**. Colors according to **Figure 2**. (**B**) Tetrapods' climatic regions are labeled according to **Figure 2**.

The online version of this article includes the following source data for figure 3:

**Source data 1.** rasters with climate regions for each studied group.

**Table 1.** Similarity of climatic regions measured by adjusted mutual information.

|  | Köppen | Tetrapods | Amphibians | Reptiles | Birds |
|---|---|---|---|---|---|
| Tetrapods | 0.44 | – | – | – | – |
| Amphibians | 0.40 | 0.66 | – | – | – |
| Reptiles | 0.45 | 0.70 | 0.58 | – | – |
| Birds | 0.45 | 0.77 | 0.61 | 0.64 | – |
| Mammals | 0.47 | 0.72 | 0.57 | 0.66 | 0.68 |

## Geographical signal in climatic regions

Historical and geographical processes may lead to species pools adapted to a given climate in some regions of the Earth but not in others. To search for this geographical signal, we first compared the distribution of the climatic conditions and species grouped within the same niche domain. A geographic mismatch between species and climate distributions would point to portions of the climatic regions that are defined by species occurring in other geographic areas. Exploring these patterns for each niche domain revealed notable geographic agreement between species and climatic conditions of the same domain (*Figure 4B* and *Appendix 1—figures 3–7*). Nevertheless, we found some

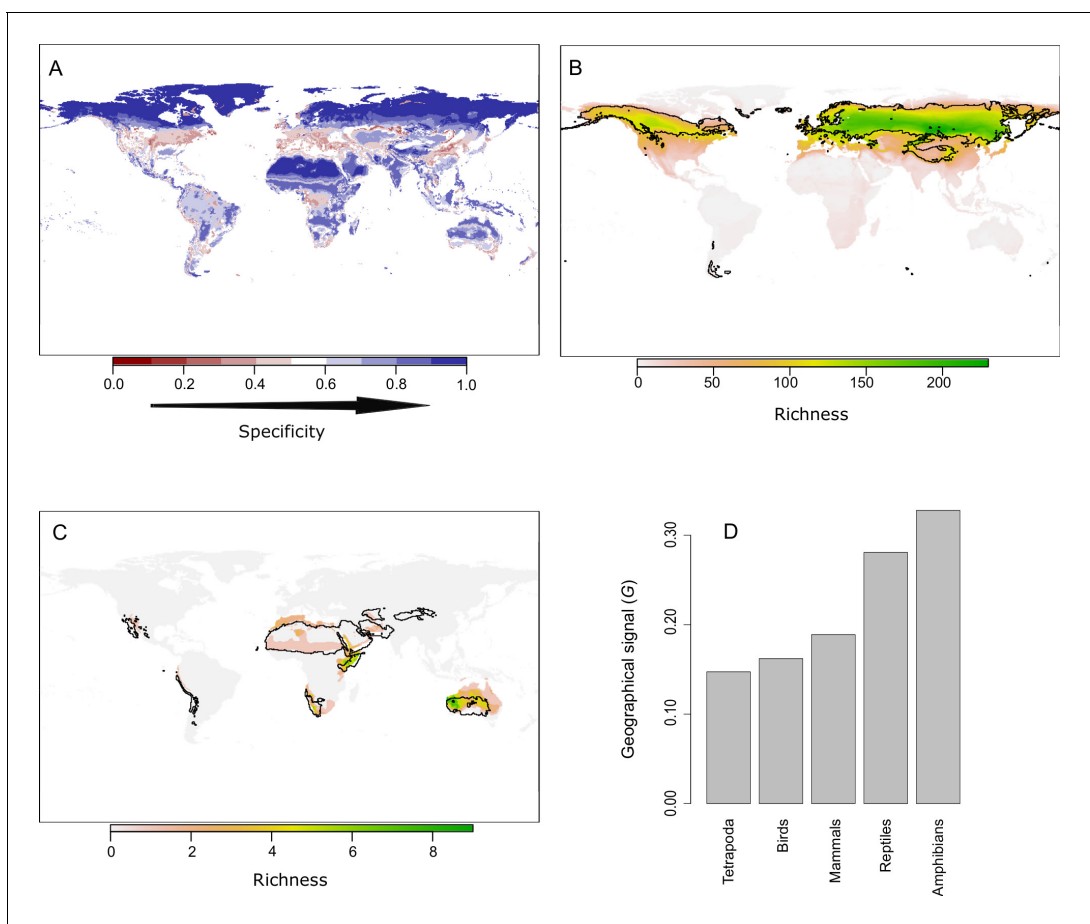

**Figure 4.** The geographic location of climatic domains and their associated species provides insights into the mechanism underlying the climatic regions. (A) Geographic projection of the specificity of climatic bins to their niche domain. (B) An example showing a bird's niche domain with a low geographical signal. The distribution of the climatic conditions (black line) and the species (colored richness values) belonging to the same niche domain was mostly congruent. (C) An example of an amphibian's niche domain showing a high geographical signal, reflected in a substantial mismatch between the distribution of climatic conditions and species belonging to the same domain. (D) A quantitative approximation of the geographical signal, ranging between 0 and 1, for the different taxonomic groups (see 'Materials and methods').

differences across groups and regions. More extreme climates showed larger mismatches between the distribution of species and climates. For instance, for all groups but reptiles, desert climate ($E^H W^L$) was mostly defined by species inhabiting Australia and to a lesser extent by species from the Namibian desert and the Horn of Africa, with few or no species inhabiting the Sahara desert (*Figure 4C* and *Appendix 1—figures 3–7*). Similarly, the northern climatic regions of amphibians and reptiles were defined by species at lower latitudes (*Appendix 1—figures 3* and *4*). Approaching the geographical signal more quantitatively (see 'Materials and methods'), we found a stronger signal for the worst dispersers, amphibians and reptiles, than for mammals and birds (*Figure 4D*), suggesting that dispersal capabilities can contribute to the geographical signal in the niche domains. Finally, all tetrapods together showed the lowest geographical signal, which suggests that, in addition to dispersion, increased evolutionary time can reduce the geographical signal.

## Discussion

We detected 16 climatic regions governing the distribution of tetrapods. Despite the substantial physiological and functional differences among the groups, some of their niche domains and climatic regions are consistent. These climatic regions also resemble Köppen's regions and correspond, in general, with extreme climates, being arid climates specially congruent across groups. These climates also presented overall low levels of transitivity, showing that species adapted to other climates have more difficulty withstanding these conditions. These results suggest that extreme climates impose strong adaptive barriers, producing a filtering of species (*Butterfield, 2015*; *Cadotte and Tucker, 2017*) even across distinctive evolutionary lineages.

Conversely, milder climatic conditions, especially temperate climates, showed the lowest statistical support and congruence across groups, as well as the highest transitivity. These results indicate that temperate climates are more difficult to classify (low support and congruence) due to the overlap in the climatic space of species pools with different climatic optima (high transitivity). Two complementary reasons can explain this ambiguity: first, while we used two variables widely recognized to shape the distributions of tetrapods, alternative variables, such as seasonal changes in energy and precipitation (*Köppen and Geiger, 1930*), may also influence species inhabiting temperate regions. Including these variables might help to further separate temperate species pools across the climatic space. Second, the climatic conditions of these domains may not prevent the colonization of species with other realized optima or preferences, which would generate the observed overlap in the climatic space (i.e., high transitivity) across milder conditions. Questions remain about the relative contribution of each process. Similarly, the exact causes and consequences of the climatic transition zones call for future investigation.

We also found some domains that were well supported but unique for each group. These differences between groups seem to relate to the particular physiological adaptations of each group. For instance, homeothermic birds and mammals defined a region of low energy, consistent with Köppen's polar climates, that reptiles and amphibians lacked. Similarly, reptiles, a group that includes several groups adapted to arid environments (*Pie et al., 2017*), defined some regions of low precipitation and high PET. Hence, our results stress that caution is needed when generalizing the climate classification to other groups of organisms, and question the validity of using plant-based classifications for studies dealing with animals.

Beyond niche domains, our results also show differences in the geographical signal across groups. That amphibians – the group with the lowest dispersal capacity – showed the highest geographical signal suggests that dispersal processes play an essential role: species that are worse dispersers have more difficulties tracking their preferred climates (*Araújo and Pearson, 2005*), limiting the colonization of disjoint areas with similar climates. Moreover, all tetrapods together showed the lowest geographical signal, suggesting that increased evolutionary time can reduce this geographic pattern. That is, an extended evolutionary time that enables the appearance of convergent adaptations to similar climates in different geographic regions (*Mazel et al., 2017*) may reduce the geographical signal in niche domains. Alternatively, tetrapods include taxa with varying dispersal strategies – such as the high active dispersal of birds and the elevated survival probabilities of reptiles to passive transoceanic dispersal – which may also influence the geographical signal. In any case, the ultimate causes and consequences of this signal require further attention. Why are some amphibians able to inhabit arid conditions in the Australian desert but not in the Sahara desert (*Appendi 1—figure 3*)?

Why can some reptiles withstand cold climates around the Himalayan mountains but not in the northern hemisphere (*Appendi 1—figure 4*)? These are some examples of the emerging fundamental questions related to the geographical signal. Answering them would require studying the adaptations that allow the species to tolerate particular climatic conditions and the abiotic, biotic, and historical differences between regions (*Moncrieff et al., 2015*) that favor these adaptations in some places but not in others. Moreover, the geographical signal idea could also help unravel idiosyncratic processes occurring at smaller scales by comparing local climate regions with global or regional ones. These examples form exciting avenues for future historical biogeographical and evolutionary studies.

Our results and data-driven methodology have the potential to bring us closer to a definition of climatic regions that represent active factors for the spatial organization of life. Nevertheless, it would be interesting to improve some aspects in future studies. First, while we used a notable number of species (about 26,000), they are taxonomically biased and only represent a small fraction of terrestrial organisms. Second, we used two climatic variables widely known to affect the distribution and diversity patterns of animals and plants in general (*Currie, 1991*; *Hawkins et al., 2003*), but other climatic variables might refine some of the least supported regions. Finally, our domains represent portions of the realized climatic niche space, which entails two sources of uncertainty. On the one hand, the estimation of realized niches depends on the data quality, scale, and treatment choices. On the other hand, the realized niche space may be influenced by historical, geographical, and biotic factors beyond pure climate (*Warren et al., 2014*; *Calatayud et al., 2019b*; *Soberón, 2007*). Hence, using fundamental rather than realized niches may also improve the accuracy of defining climatic regions. At the current pace of biological data accumulation and computational development, it is reasonable to expect that some of these limitations will soon be overcome. Meanwhile, the considerable congruence of several climatic regions across the studied groups and Köppen's system provides confidence in their robustness. Hence, it is likely that using better data would not produce regions substantially different from those presented here.

Regardless of how generalizable the results are, the niche domains and their associated species pools and climatic regions can be used as a basis for ecological, evolutionary, and conservation studies concerning tetrapods. Some of the many questions that the results reported here (data available in source data of *Figures 2* and *3*) can help to answer include: Are the species belonging to different niche domains similarly conserved or protected, or both (*Hanson et al., 2020*)? To what extent do the differences between the four classes of tetrapods reflect phylogenetic and functional differences, and are such differences to be found in other taxonomic groups? Is the adaptation to niche domains evolutionarily constrained? Do diversification, extinction, or speciation rates differ among the species associated with different domains? Moreover, combining niche domains with bioregions based on pools of species (*Bernardo-Madrid et al., 2019*) or lineages (*Holt et al., 2013*) can also help answer several relevant questions: Are introductions more common between bioregions with the same climate types? To what extent does the current climate govern bioregions? Do ecological and evolutionary processes differ between different bioregions with same climate types (*Moncrieff et al., 2015*)?

In conclusion, our data-driven climate classification reveals major climatic boundaries organizing the distribution of tetrapods on Earth. Questions remain regarding the mechanism underlying the differences between groups in the climatic regions and the geographical signal. Nevertheless, these differences across groups highlight the need for classifications of climates specific to each group to study their ecology, evolution, or conservation.

## Materials and methods

### Data

We obtained the distribution ranges of mammals and amphibians from the *IUCN, 2015*, of birds from *BirdLife, 2015*, and of reptiles from *Roll et al., 2017*. We included only the native range of terrestrial species in the analyses in all instances. In the case of birds, we only used the breeding ranges. Moreover, since there is a higher uncertainty when determining the realized niches of narrow-ranging species (*Lehmann et al., 2002*), we arbitrarily removed the species whose ranges were

less than five grid cells of 0.5°. After this cleaning of the data, we used 3657 amphibians, 7204 reptiles, 4574 mammals, and 10,684 birds, for a total of 26,119 tetrapod species.

We approximated the species' Grinnellian niches (*Soberón, 2007*) with two climatic variables that represent energy and water inputs. While we could have used several other variables, we chose energy and water as they best explain climatic effects on species distributions (*Hawkins et al., 2003*). As surrogates for energy and water inputs, we used mean annual PET and AP, respectively. Both variables have been shown to be important factors for tetrapod species distributions (*Currie, 1991*; *Tingley et al., 2009*; *Gouveia et al., 2014*). Moreover, they have also been used in previous climate classifications (*Thornthwaite, 1948*) and are regularly used to derive other drivers of species distributions such as the UNEP aridity index (*Unep and Thomas, 1992*; *Fuller et al., 2016*). We obtained PET from *Trabucco and Zomer, 2009* and AP from *Fick and Hijmans, 2017*, both at a 0.08° resolution. Finally, we obtained Köppen's climatic regions from *Kottek et al., 2006* and *Rubel et al., 2017*.

## Niche characterization

We characterized the realized climatic niche of each species using an approach similar to the one proposed in *Broennimann et al., 2012*. We divided the climatic space formed by PET and AP into bins and calculated the proportion of occurrences a given species has in each climatic bin. Both the shape of the divisions and the number of divisions of each climatic axis affect the result. For instance, dividing the axis into regular intervals can destroy critical information if the climatic values more important for the species distributions are skewed toward any extreme of the distribution or if the climatic values are represented non-uniformly across the globe (as for AP, *Appendix 1—figure 8*). Similarly, the grain size to divide the climatic space may affect subsequent results (*Levin, 1992*; *Daru et al., 2020*). Dividing the space into too few intervals destroys information, whereas using too many divisions can generate niche domains with only a few species. To overcome the first issue, we divided the axes into quantiles based on the distribution of climatic values across the Earth. By doing so, we obtained an almost uniformly divided PET axis (*Appendix 1—figure 8*). Contrarily, the number of divisions of the AP axis was skewed toward low values, which resulted in a higher resolution over the presumably more relevant low-precipitation conditions (*Appendix 1—figure 8*). To solve the second issue, we selected the lowest number of divisions that maximized the gain in information (see Appendix 1). The optimal number of axis divisions was 17 in all cases but amphibians, where it was 18.

Next we accounted for potential commission errors, which may affect the estimated climates a species experiences. Specifically, range maps can overestimate the area occupied by a species, which directly influences the niche characterization (*Rondinini et al., 2006*). Because range maps typically represent the species' maximum geographical extent (*Rondinini et al., 2006*; *La Sorte and Hawkins, 2007*), extracting the climatic values that a species range covers from a well-fitting 0.08° high-resolution climatic raster can reduce commission errors at the species range's borders. But with many pixels inside the species range, the noise from extreme and unrepresentative climatic values can increase the error (*La Sorte and Hawkins, 2007*). Conversely, extracting climatic values from a coarser raster can reduce the commission errors inside a range by averaging out extreme values at the cost of increasing commission errors over the borders. To alleviate the effects of these potential errors, we first extracted the climatic values from the high-resolution rasters (0.08°). Then, we computed the average climatic values among selected raster pixels located within cells of 0.5°. In this way, we reduced the effects of commission errors both at the borders of and inside species ranges. Finally, mean climatic values may not accurately represent the cells' climates when there is high climate variability or the values are non-normally distributed. Comparing results obtained from different cell sizes is an indirect way to asses the influence of distorted mean values. Instead, we chose to directly explore the effects of high climatic variability and non-normal distributions with non-parametric bootstrap analysis: we resampled climatic values within 0.5° cells with replacement (see below).

## Niche domains and climatic regions identification

We employed a network community detection approach to identify the niche domains and the species mainly associated with them. For each group of species, we first generated a weighted bipartite

network where species and climatic bins formed the disjoint sets of nodes, and the proportion of occurrences of species in intervals of the climatic values corresponding to the climatic bins formed the weighted links. To identify the niche domains, we used the hierarchical version of the community detection algorithm known as Infomap (*Rosvall and Bergstrom, 2008*). Infomap capitalizes on the minimum description length principle of information theory, which equates finding regularities and compression: the model that finds most regularities in a given set of data can compress the data the most (*Rissanen, 1978*). In our case, modules of highly interconnected climatic bins and species form the regularities, and describing the network with an optimal set of communities corresponds to minimizing the description length (*Bernardo-Madrid et al., 2019*; *Rosvall and Bergstrom, 2008*). Among the many community-detection algorithms available, we used Infomap because it can find hierarchically nested communities and is known for its high performance (*Lancichinetti and Fortunato, 2009*), also for regionalizations (*Bernardo-Madrid et al., 2019*; *Bloomfield et al., 2018*; *Vilhena and Antonelli, 2015*). We ran the algorithm 1000 times, selecting the network partition with the best quality.

To consider the uncertainty associated with both the species ranges and the community detection, we conducted a bootstrap analysis. For each species, we resampled with replacement from the distribution of climatic values within species ranges at a resolution of 0.08°. We averaged climatic values occurring within 0.5° cells and calculated the proportion of occurrences in each climatic bin. With resampled data from all species, we generated a bootstrapped network and clustered it with Infomap 1000 times. We repeated this procedure for 100 bootstrap networks and followed the community-stability approach proposed in *Calatayud et al., 2019a* to calculate the niche domains' support. For each identified domain, we calculated the proportion of bootstrap networks with a domain more similar to Jaccard index 0.5 (*Calatayud et al., 2019a*).

With obtained niche domains, we detected the climatic regions by identifying areas across the Earth's surface that hold the climatic conditions grouped within each niche domain. Finally, to compare climatic regions across tetrapod groups and with Köppen's classification, we calculated the AMI. AMI measures the mutual information between two partitions, the classifications of raster pixels into climatic regions in this case, correcting for the similarity between partitions that are just due to chance (*Vinh et al., 2010*). The index is 1 when the partitions are equal and tends to 0 otherwise.

## Climatic transition zones

The joint classification of climatic bins and species into domains $D$ allowed us to calculate the bins' specificity. Though species belong to single domains, typically together with the bins to which they have many and strong links, they may also have links to bins in other domains. Bins that contain species from different domains have low specificity and form a transition zone between domains (*Figure 1*; *Calatayud et al., 2019b*; *Bernardo-Madrid et al., 2019*). We calculated the specificity $S_i^D$ of a climatic bin $i$ in domain $D$ as the sum of link weights $w_{i,j}$ from bin $i$ to species $j$ assigned to the same domain as the bin, divided by the sum of link weights from bin $i$ to all species $j$ such that

$$S_i^D = \frac{\sum_{j \in D} w_{i,j}}{\sum_j w_{i,j}} \text{for} i \in D. \tag{1}$$

This index is 1 when the bin connects only to species in the same domain and tends to 0 otherwise.

We projected the specificity values into the geographic space. To all geographical raster cells $q$ with the climate of climatic bin $i$, which we call raster cell set $Q_i$, we assigned bin $i$'s specificity. The projected specificity is $S_q^P$.

Finally, we explored the relationship between the average $S^D$ and bootstrap support. We fitted a logistic GLMM of bootstrap p-values as a function of the mean $S^D$ and the taxonomic group. We used the R (*R Development Core Team, 2018*) package lme4 (*Bates et al., 2015*) with the mean $S^D$ as a fixed term and the taxonomic group as a random intercept term.

## Geographical signal

To quantify the geographical signal, we compared the geographically projected specificity $S^P$ with the actual specificity based on the pool of species that co-occur geographically. Large differences

between the species co-occurring in the climatic and geographic spaces indicate a strong geographic signal. For instance, when species assigned to the same domain co-occur only in a portion of the corresponding geographical space, geographical areas with mismatching species contribute to a strong geographical signal. Areas that host most of the species associated with a niche domain have a higher actual specificity than the projected indicates. Areas that are not, or only scarcely, colonized by these species have a lower actual specificity than the projected indicates.

Using *Equation 1*, we calculated the actual specificity of a geographical raster cell $q$, whose corresponding climatic bin $i$ is in domain $D$ ($q \in Q_i$ and $i \in D$), as the ratio between the sum of link weights from bin $i$ to species in raster cell $q$ that belong to its associated domain and the total link weight from bin $i$ to all species in $q$,

$$S_q^A = \frac{\sum_{j \in D,q} w_{i,j}}{\sum_{j \in q} w_{i,j}} \text{ for } q \in Q\%_i \text{ and } i \in D. \tag{2}$$

To calculate the geographical signal $G$, we averaged the absolute difference between the projected and actual specificity of each climatic raster cell $q$ at 0.5° resolution,

$$G = \frac{1}{N} \sum_{q=1}^{N} |S_q^A - S_q^P|, \tag{3}$$

where $N$ is the total number of raster cells. This index is 0 when there is no geographical signal and tends to 1 for high signals.

## Acknowledgements

We are thankful to Andrea Briega and Miguel Á. Rodríguez for discussion on early ideas. We are very grateful to Fernanda Alves-Martins, Rafaél Molina-Venegas, Cristina Roquillo, Rubén Bernardo-Madrid, Ana G Garcia, Daniel Fuentes, and Marina Coca for critical reviews. JC was supported by the Carl Tryggers Foundation for Scientific Research (CTS 16:384), AR by the Olle Engkvist ByggmÄ¤stare Foundation, and MR by the Swedish Research Council, grant 2016–00796.

## Additional information

### Funding

| Funder | Grant reference number | Author |
| --- | --- | --- |
| Carl Tryggers Stiftelse för Vetenskaplig Forskning | | Joaquín Calatayud |
| Stiftelsen Olle Engkvist Byggmästare | | Martin Rosvall |
| Svenska Forskningsrådet Formas | | Magnus Neuman<br>Martin Rosvall |
| Swedish Research Council | 2016–00796 | Martin Rosvall |

The funders had no role in study design, data collection and interpretation, or the decision to submit the work for publication.

### Author contributions

Joaquín Calatayud, Conceptualization, Resources, Data curation, Software, Formal analysis, Supervision, Validation, Investigation, Visualization, Methodology, Writing - original draft, Writing - review and editing; Magnus Neuman, Conceptualization, Resources, Software, Formal analysis, Supervision, Validation, Investigation, Visualization, Methodology, Writing - review and editing; Alexis Rojas, Validation, Investigation, Methodology, Writing - review and editing; Anton Eriksson, Formal analysis, Validation, Visualization, Methodology, Writing - review and editing; Martin Rosvall, Resources, Software, Formal analysis, Supervision, Funding acquisition, Validation, Investigation, Visualization, Methodology, Writing - review and editing

## Author ORCIDs

Joaquín Calatayud (iD) https://orcid.org/0000-0001-9056-4149
Alexis Rojas (iD) https://orcid.org/0000-0002-1063-9102

## Decision letter and Author response

Decision letter https://doi.org/10.7554/eLife.58397.sa1
Author response https://doi.org/10.7554/eLife.58397.sa2

## Additional files

### Supplementary files
- Transparent reporting form

### Data availability

Data used is freely available in the sources detailed in the Methods section. Data generated in this study is provided as source data associated to Figures 2 and 3.

The following previously published datasets were used:

| Author(s) | Year | Dataset title | Dataset URL | Database and Identifier |
|---|---|---|---|---|
| Roll U, Feldman A, Novosolov M, Allison A, Bauer AM, Bernard R, Böhm M, Castro-Herrera F, Chirio L, Collen B | 2017 | The global distribution of tetrapods reveals a need for targeted reptile conservation | https://doi.org/10.5061/dryad.83s7k | Dryad Digital Repository, 10.5061/dryad.83s7k |

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

## Appendix 1

### Selecting the number of bins

Dividing the climatic space into an optimal number of bins is critical because it can affect subsequent classifications. On the one hand, an insufficient number of bins may destroy information. For instance, in the extreme case of using only one bin, all the species will belong to the same climatic niche. On the other hand, an excess of bins may lead to niche domains with only a few specialist species, which are not suitable for a global climate classification. The optimal value is in between these extremes. Intuitively, the differences in the species' climatic niches increase with the number of bins. Yet, when the bins are enough to capture the main differences among species' niches, increasing the number of bins will not increase these differences. In other words, when the number of bins is sufficient to describe the main information of the system, more bins will not significantly increase the information contained in the species niches. Therefore, when the number of bins already captures the main information, the slope of the relationship between the differences in species' niches and the number of bins becomes less pronounced.

To measure the average differences between the niches, we used the well-known Jensen–Shannon divergence (JSD). Assuming that each species niche is a probability distribution $P_i$ across the discretized climatic niche space,

$$JSD = H\left(\sum_{i=1}^{n} \frac{P_i}{n}\right) - \sum_{i=1}^{n} \frac{H(P_i)}{n}, \tag{4}$$

where $H(P)$ is the Shannon entropy for the species niche $P$ and $n$ is the number of species. The JSD is bounded between 0, when the probability distributions are equal, and $\log_2(n)$, when the distributions are completely dissimilar. To make results comparable, we normalized JSD by $\log_2(n)$.

We discretized the climatic space by dividing PET and AP in quantiles, ranging from 5 to 50 and increasing in steps of 5. We used the same number $n$ of quantiles in both variables, so the 2D climatic space has $n^2$ bins. For each division, we computed the proportion of observations of each species in each bin and calculated the JSD using all species. Finally, we fitted a piecewise regression of the increments of JSD ($\Delta JSD$) as a function of the number of divisions. We found that the optimal number of divisions was 17 in all cases but amphibians, where it was 18 (*Appendi 1—figure 9*).

## Supporting figures

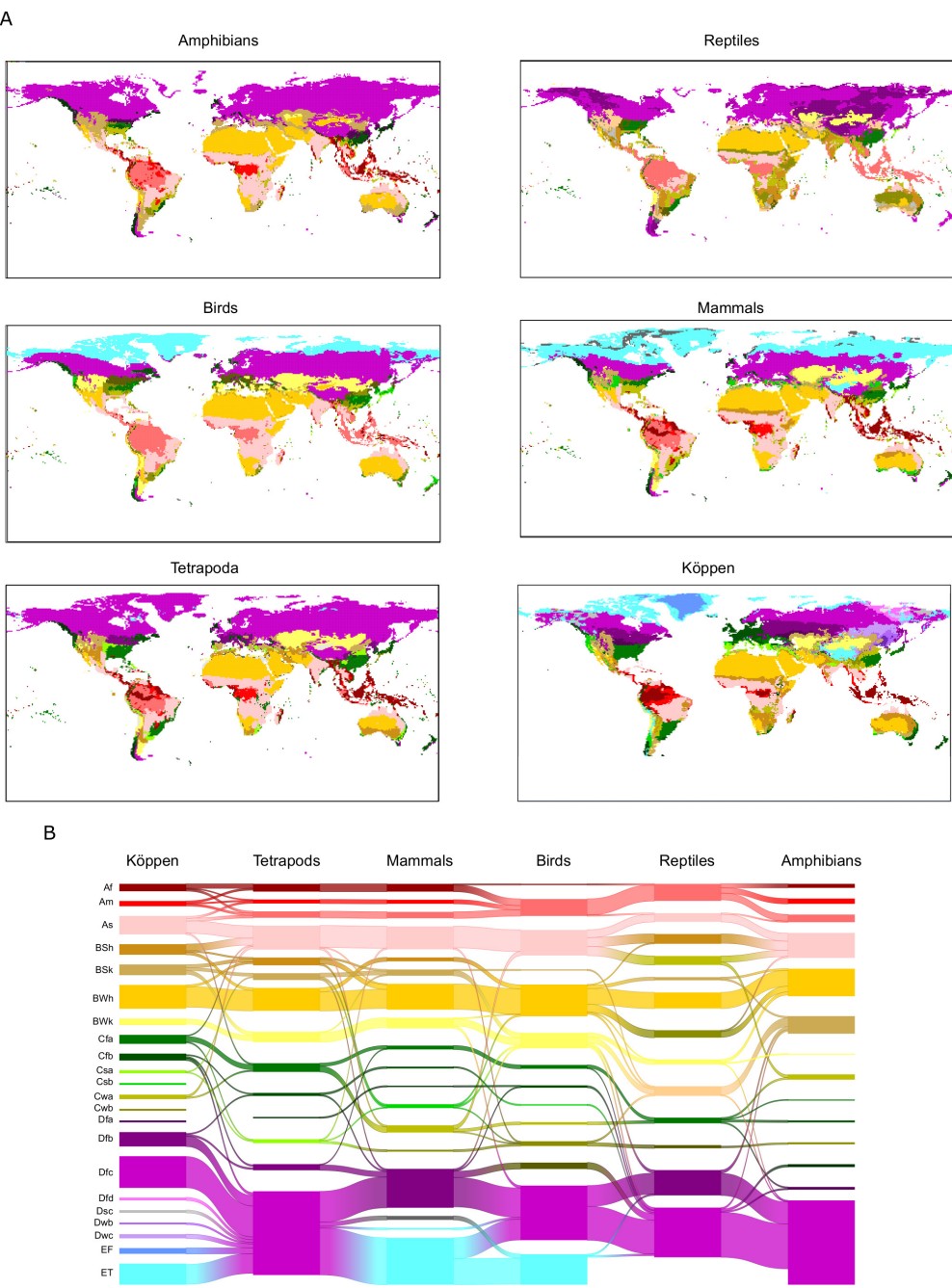

**Appendix 1—figure 1.** Tetrapod groups and Köppen's climatic regions. (**A**) Geographic location of tetrapods' niche domains and Köppen's climatic regions. (**B**) Alluvial diagram comparing the climatic regions between classifications. In the stacks of boxes, one for each classification, a box represents a climatic region, and the height of the box is proportional to the number of raster cells of the climatic region. Streamlines connecting boxes in different stacks depict the number of raster cells shared between climatic regions of different classifications. Colors according to *Figure 2*.

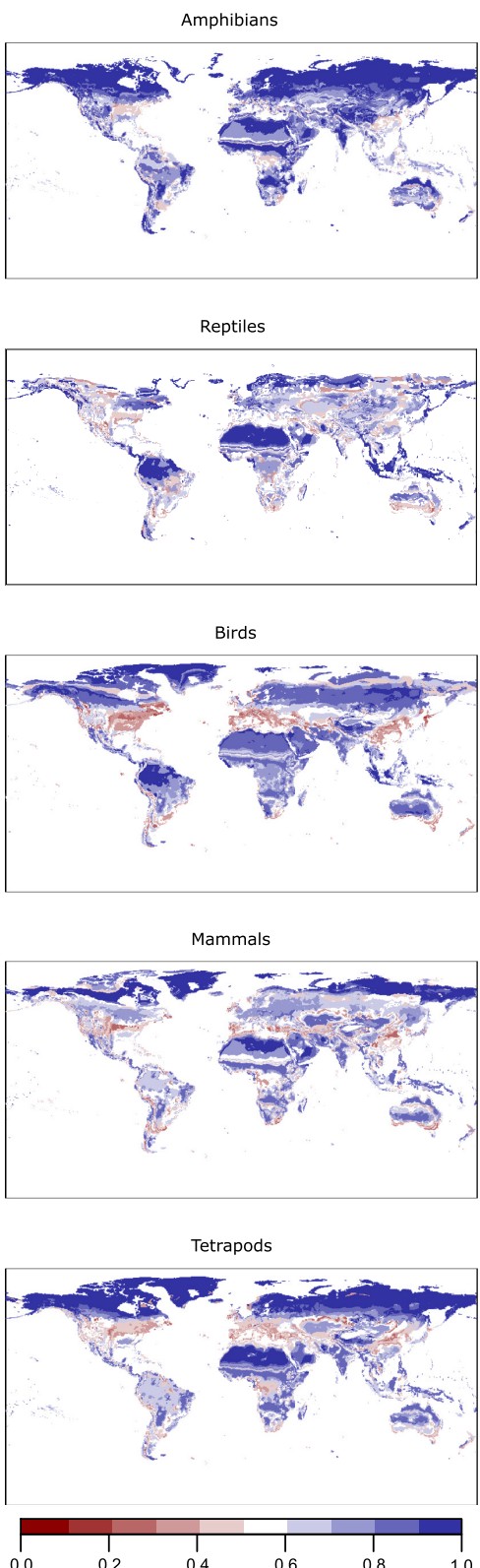

**Appendix 1—figure 2.** Geographic projection of the specificity of the climatic bins ($S^P$) for the studied groups.

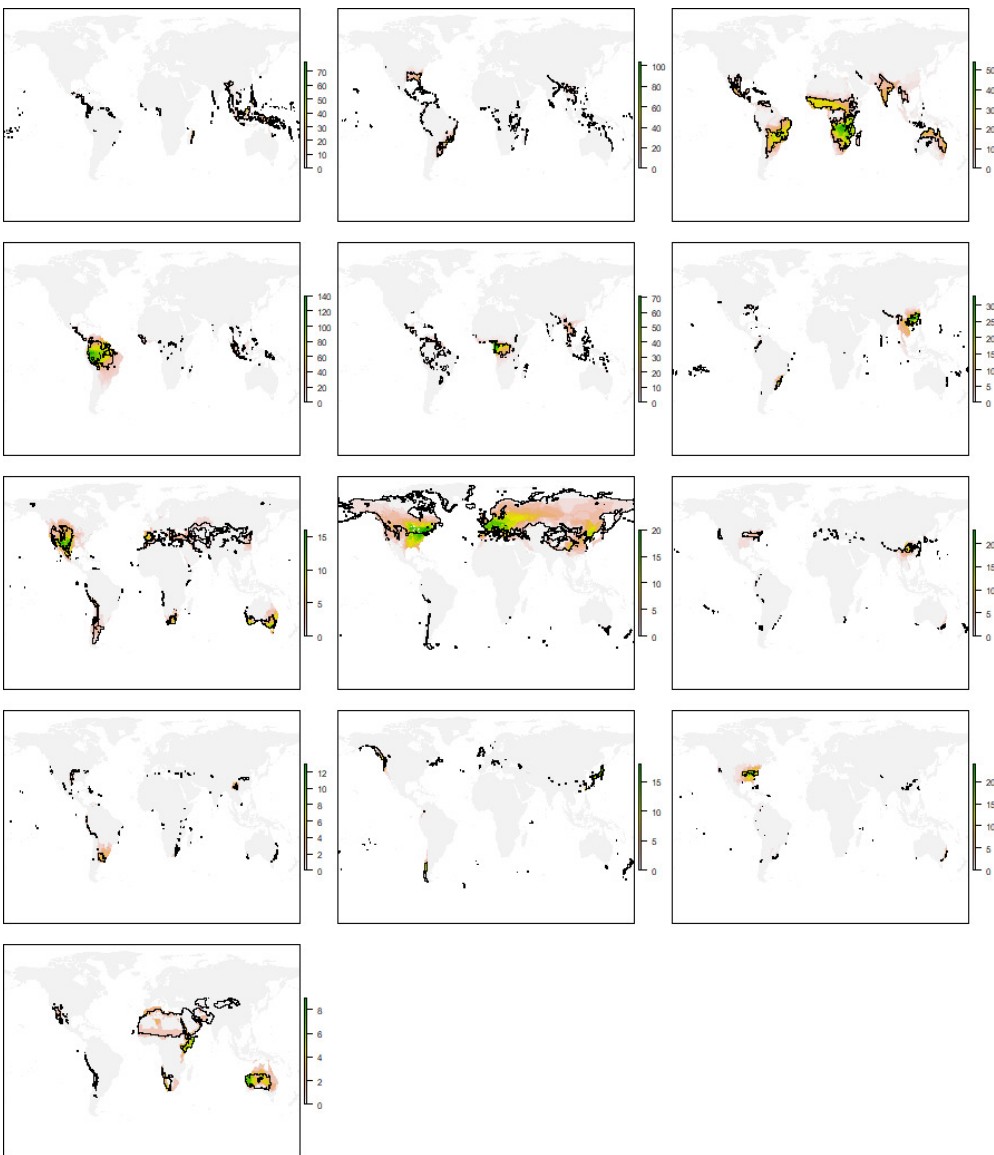

**Appendix 1—figure 3.** Distribution of the climatic conditions (black line) and the species (colored richness values) belonging to the same niche domains of amphibians.

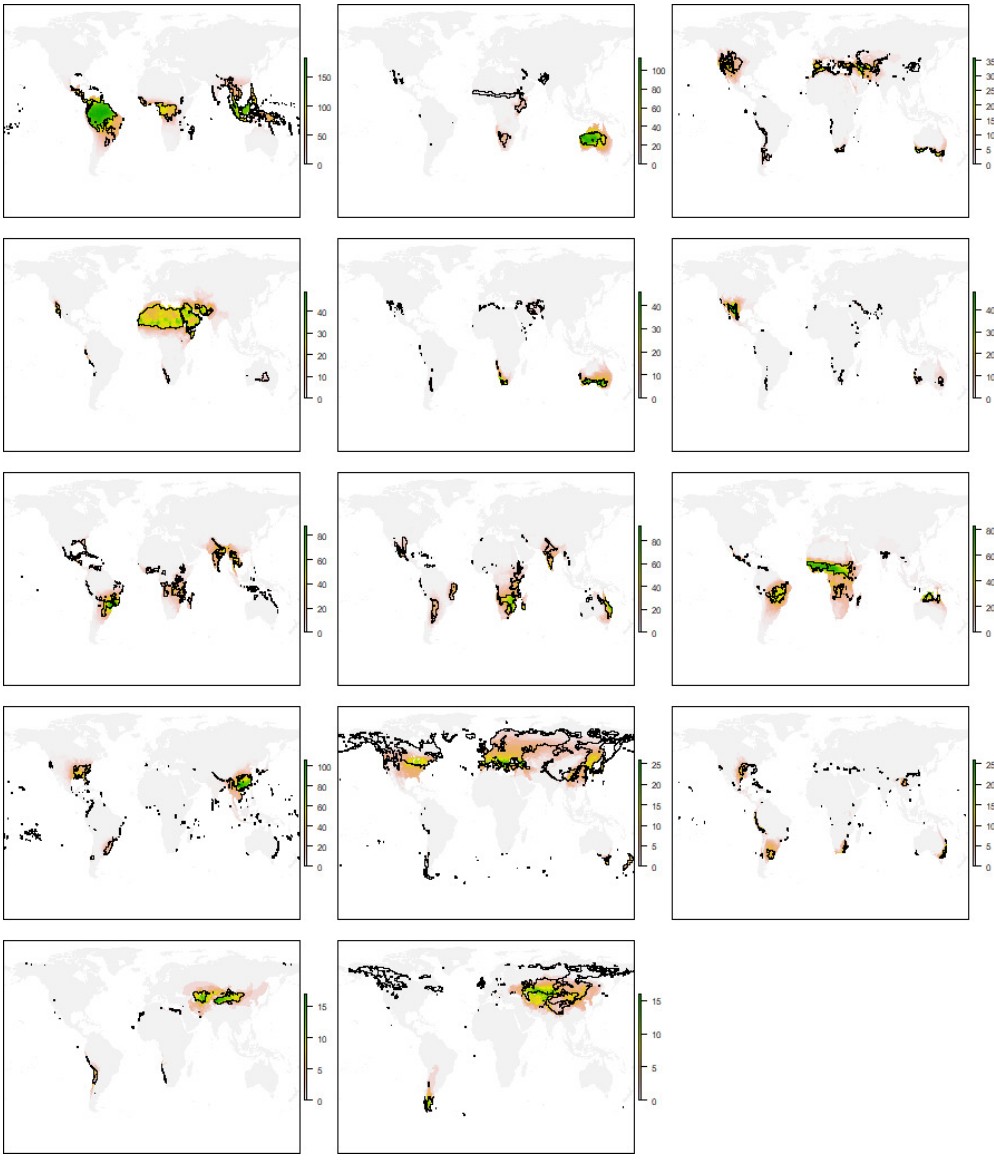

**Appendix 1—figure 4.** Distribution of the climatic conditions (black line) and the species (colored richness values) belonging to the same niche domains of reptiles.

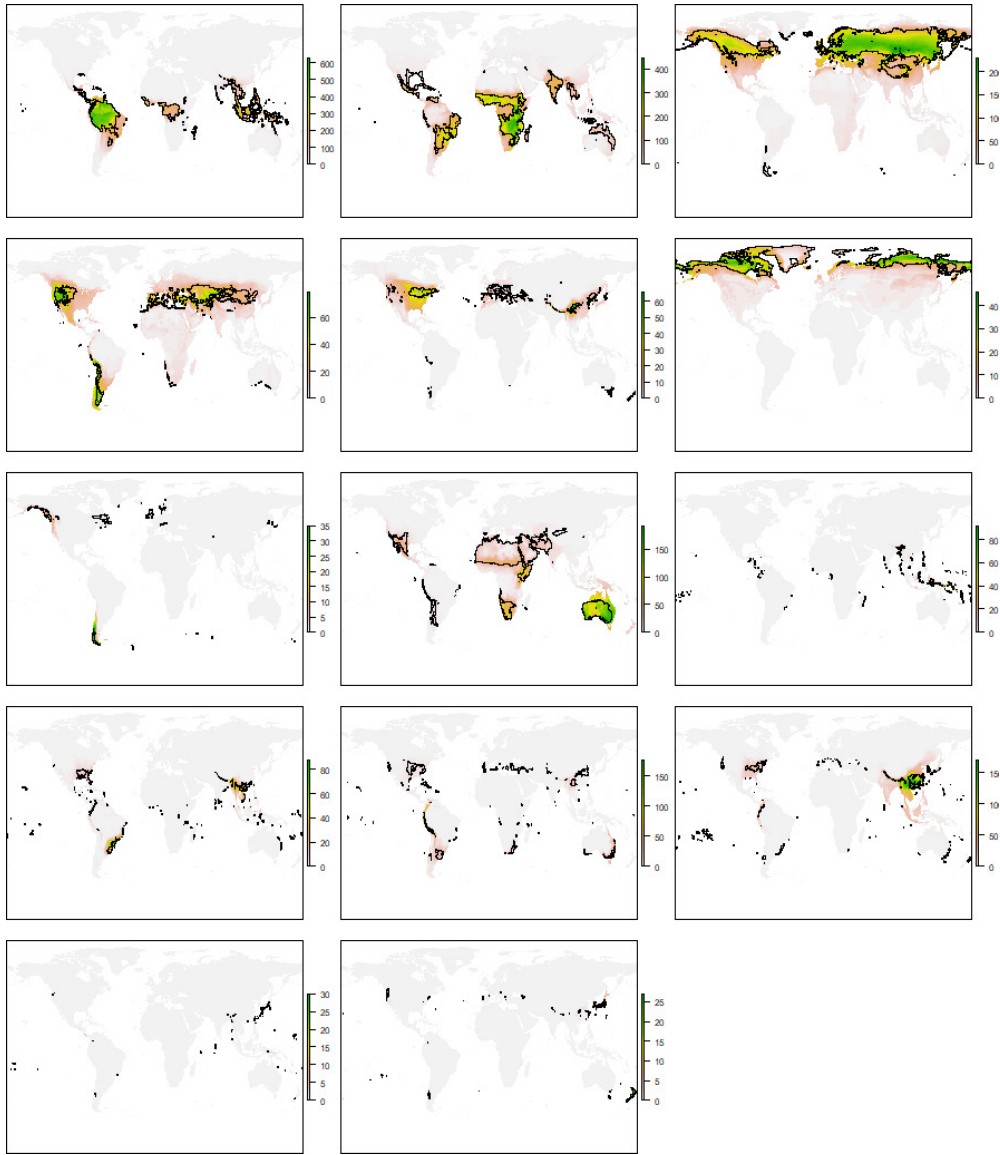

**Appendix 1—figure 5.** Distribution of the climatic conditions (black line) and the species (colored richness values) belonging to the same niche domains of birds.

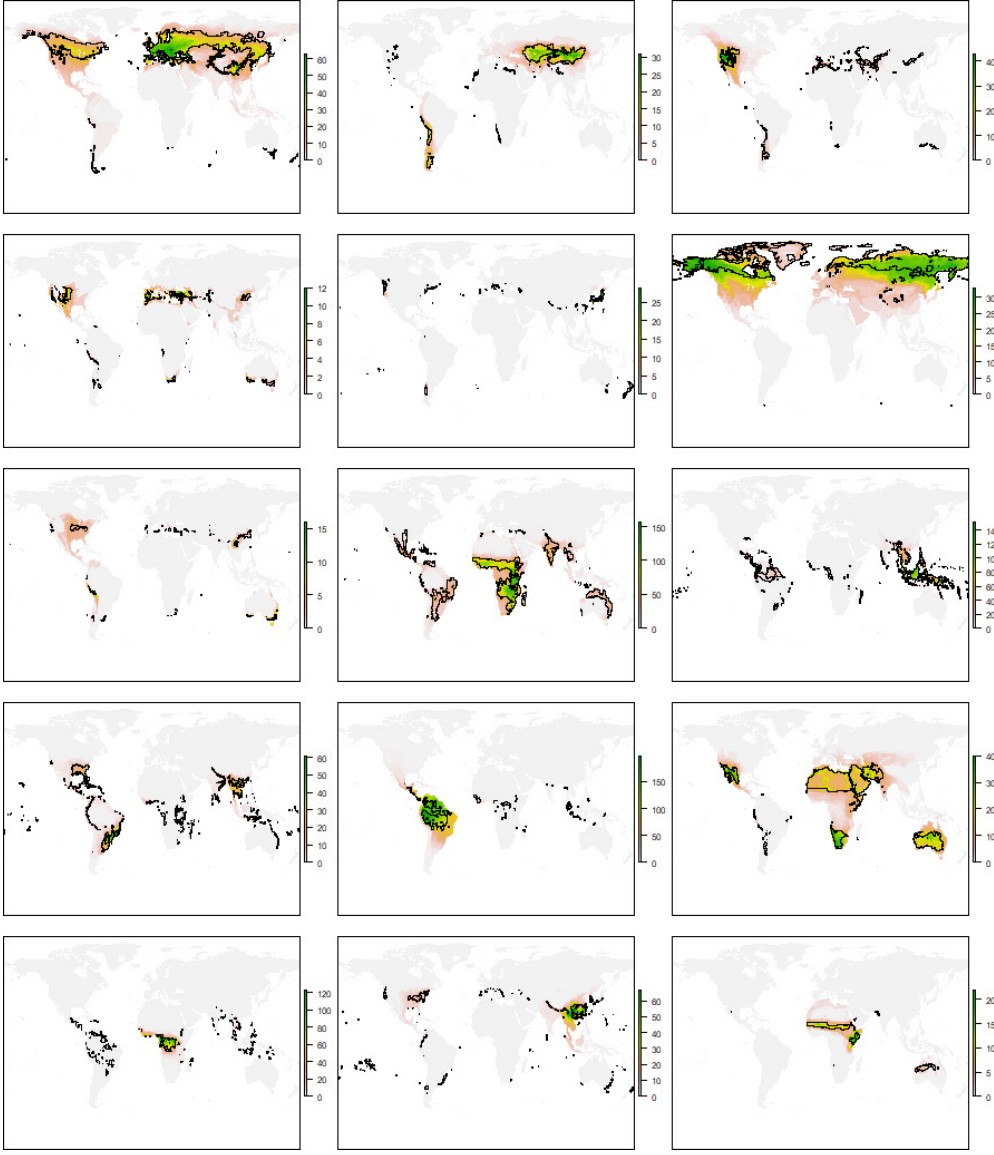

**Appendix 1—figure 6.** Distribution of the climatic conditions (black line) and the species (colored richness values) belonging to the same niche domains of mammals.

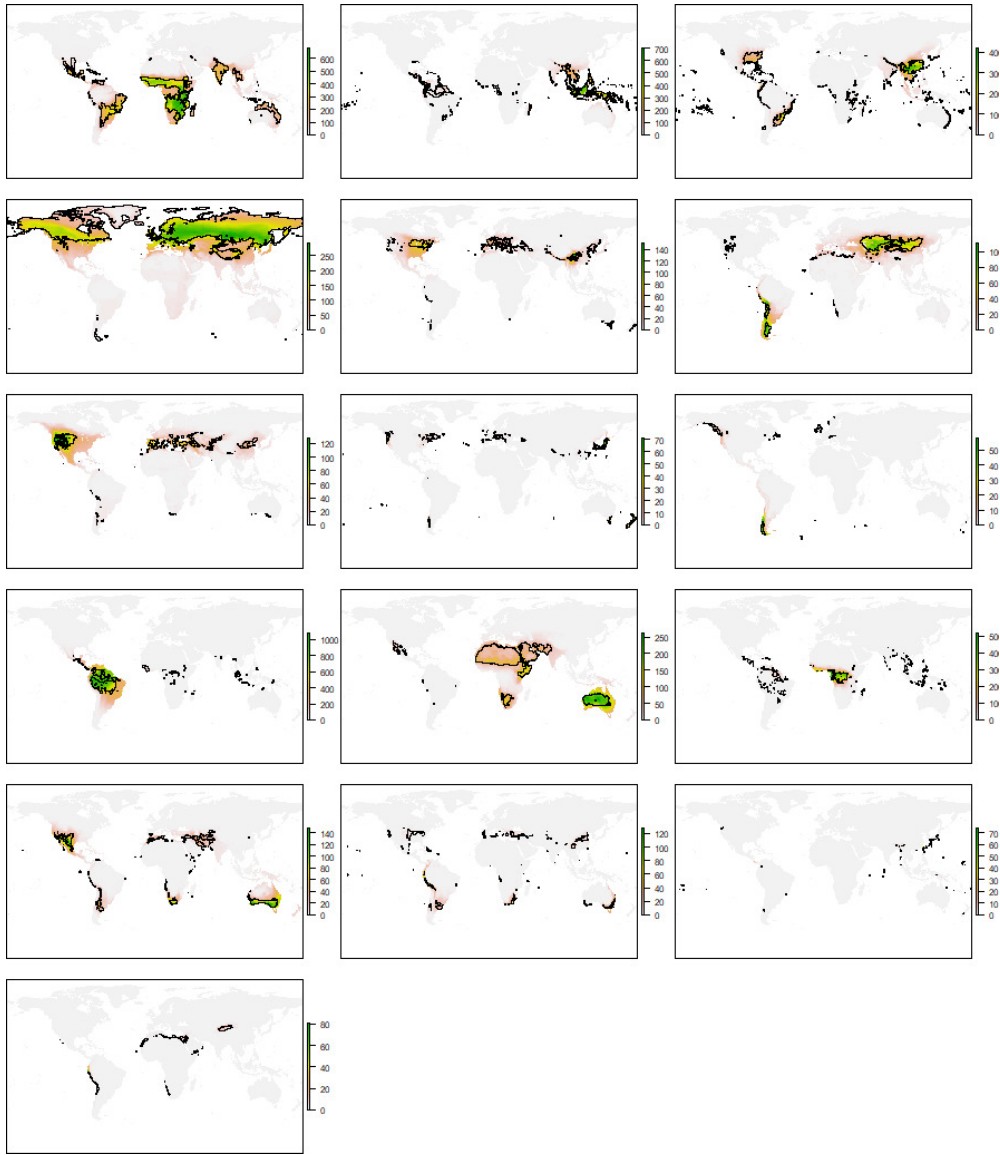

**Appendix 1—figure 7.** Distribution of the climatic conditions (black line) and the species (colored richness values) belonging to the same niche domains of tetrapods.

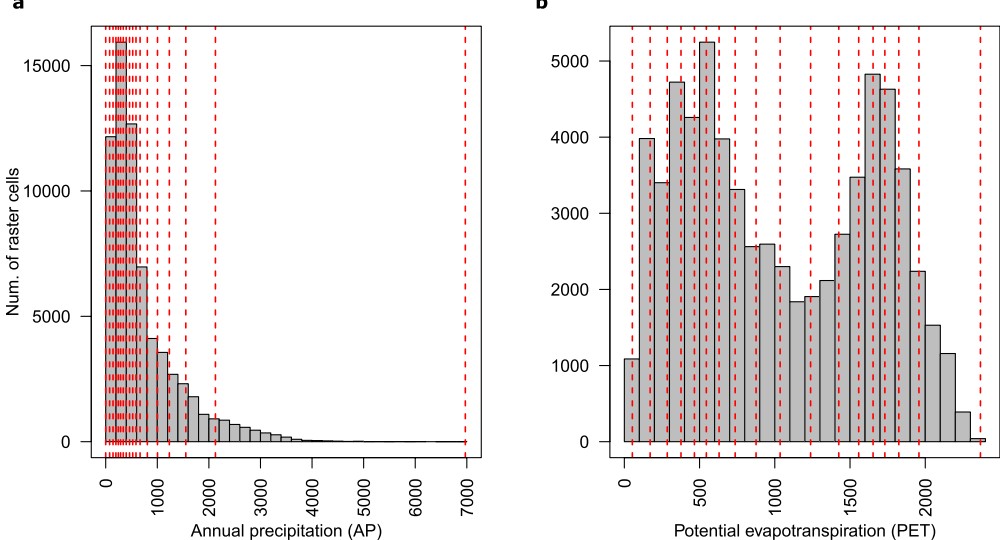

**Appendix 1—figure 8.** Distribution of (**a**) annual precipitation and (**b**) potential evapotranspiration values. Seventeen divisions of each climatic variable are presented with red dotted lines.

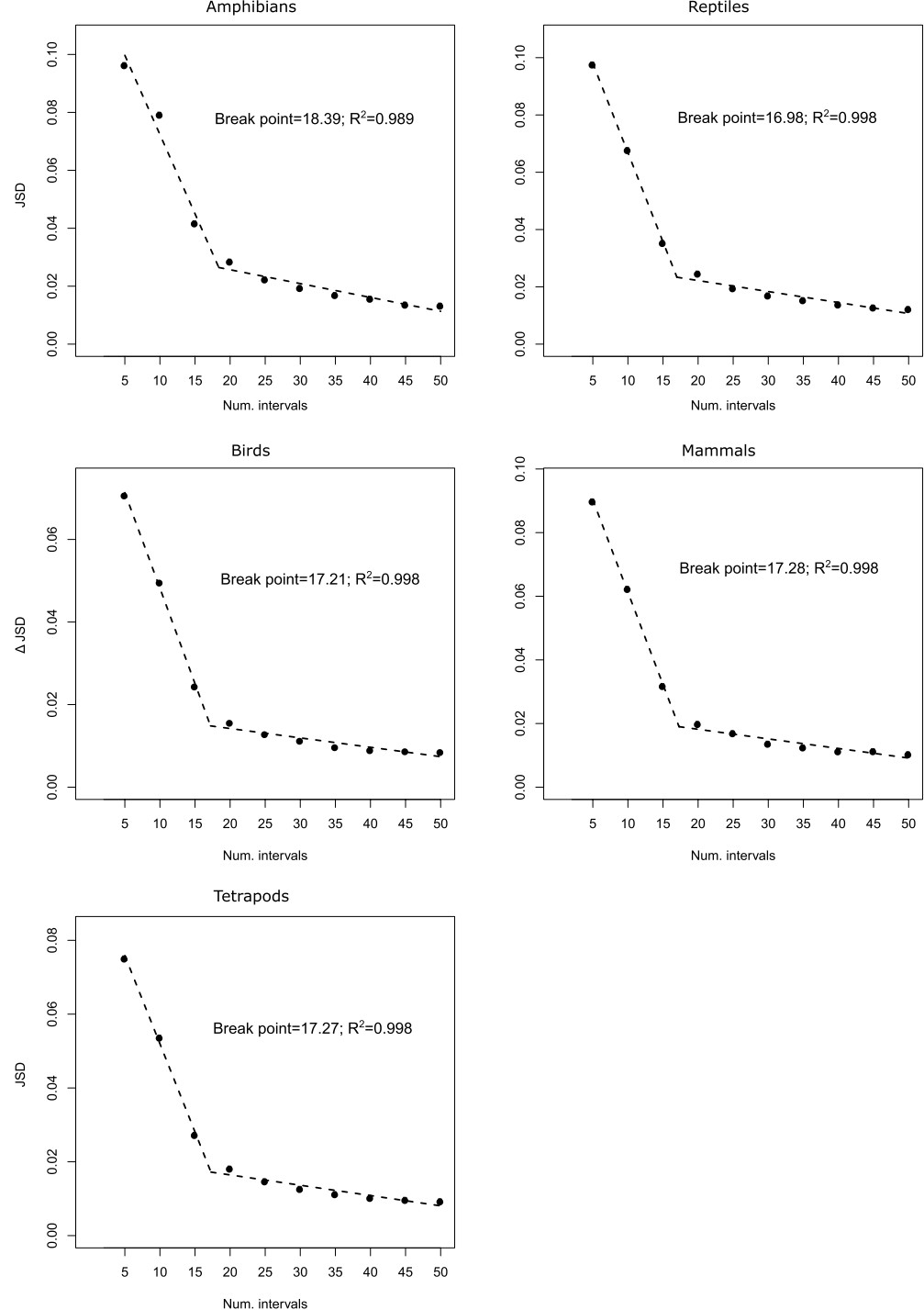

**Appendix 1—figure 9.** Predictions of piecewise regression of the increment in Jensen–Shannon divergence ($\Delta JSD$) as a function of the number of divisions in the variables defining the climatic space of all studied groups. The breakpoints and the coefficient of determinations are also provided.

