## [Decision Letter]

**Acceptance summary:**

The climatic determinants shaping animal distributions are largely unknown. Network modelling of realized niche domains of 26,000 vertebrates, accounting for the climatic conditions a species experiences within its range, reveals new classification of global climate regimes.

**Decision letter after peer review:**

Thank you for submitting your article "Regularities in species' niches reveal the world's climate regions" for consideration by *eLife*. Your article has been reviewed by three peer reviewers, including Barnabas Daru as the Guest Reviewing Editor and Reviewer #3, and the evaluation has been overseen by Christian Rutz as the Senior Editor. The following individuals involved in the review of your submission have agreed to reveal their identity: Kevin Cazelles (Reviewer #1); Serban Proches (Reviewer #2).

The reviewers have seen each other's reviews as part of a post-review consultation session. Since there were no disagreements, we have decided to append their separate original reports to this decision letter. Please address the comments as completely as possible in a revised manuscript.

Since revisions are required before the work can be published, we would like to draw your attention to changes in our revision policy that we have made in response to COVID-19 (https://elifesciences.org/articles/57162). First, because many researchers have temporarily lost access to the labs, we will give authors as much time as they need to submit revised manuscripts. We are also offering, if you choose, to post the manuscript to bioRxiv (if it is not already there) along with this decision letter and a formal designation that the manuscript is "in revision at *eLife*". Please let us know if you would like to pursue this option. (If your work is more suitable for medRxiv, you will need to post the preprint yourself, as the mechanisms for us to do so are still in development.)

Reviewer #1:

My comments are meant to be constructive, and I hope they will be helpful as you revise your manuscript.

Overall opinion

In this manuscript, Calatayud et al. use a network-based approach to identify climatic regions based on tetrapods occurrence. With these in hand, they assess their congruence with the regions of the Köppen-Geiger climate classification (based on climate and plant distributions) and they also assess their specificity. Overall, I think that this study has a lot of merit and I truly value the work done. However, the current version of manuscript requires some work before being publishable. I have several major concerns that I develop below.

Lack of clarity

The lack of clarity is twofold. First, some sentences are just not clear and need to be rephrased. Below are three examples that all concern the description of the method:

– Subsection “Climatic transition zones” I had to read this several times to understand the meaning of it. A formal definition of D is missing (what are the species included in it).

– Subsection “Geographical signal” I think the equations are valid for 1 domain, but this is not made clear (e.g. how what makes index $i$ disappear from (2) to (1))

– The text in Appendix 1 is hard to read.

Because of this lack of clarity, I am not 100% sure about what has actually been done. It might just be a matter of being careful with the annotations and carefully detailing all the steps, so it might not be much work, but this needs to be done.

Second, some details are missing in critical parts. For example:

– "Using a hierarchical network clustering algorithm (Rosvall and Bergstrom, 2008, 2011)". This is a critical step and the procedure is published, but I think we need more details to understand what is done, i.e. the rational of the algorithm and why it is suited for this analysis. Also, I don't understand where the assessment of the domain's robustness is done in the Appendix.

– AMI is not (yet) frequently used in ecology so I would remind the reader what is it otherwise the reader may not understand the meaning of a AMI of 0.7 (e.g.).

My point being that the right level of methodological is not always provided. A few more sentences may do the jobs.

Why this is not all trivial?

Full disclaimer, I don't think the results are trivial. But at first sight, one may think so.

After my first read of the manuscript, I thought that the two following assertions of the Introduction were contradictory

– "abiotic conditions determine species ranges"

– "However, the fact that plant species are good indicators of general climatic conditions does not necessarily imply that such conditions shape the distribution of other organisms in the same manner"

If "abiotic conditions determine species ranges" and if the reference classification is based on plant distributions, given the variety of ecological links between plants and tetrapodes, then the distribution of tetrapodes communities (assuming there are properly defined) and the reference distribution (Köppen) should be very similar. And actually they are similar as it is showed in Figure 3. But after thinking more about this I convinced myself otherwise, partly thanks to the manuscript, but not only. And this is the issue: the authors should better explain why this is not all trivial in the Introduction, they should explain why one could expect significant discrepancies between the two distributions.

Impact statement

Importantly enough, I don't think that the impact statement : "Similarities in climatic niches of terrestrial vertebrates indicate the Earth's climate regions, which substantially differ from previous plant-based climate classifications." is a fair depiction of the results and it is actually contradictory with, inter alia, the results presented in Figure 3 (the caption of which reads "Tetrapoda groups and Köppen's climatic regions are largely congruent"). Unless I have miss something.

Discussion

I must say that the Discussion section includes various bold statements that the authors, for instance:

"Our results bring us closer to a definition of climatic regions that represent active factors for the organization and evolution of life."

Quite frankly, after reading the manuscript, I am not entirely sure why. I would rather discourage the authors from making such statements.

Reviewer #2:

This is a brilliant piece of work, filling in a massive gap in biogeography. I think for a first shot, including just one measure of energy and one for water is perfectly fine. The writing is good and the logic mostly consistent, with very slight lapses in the discussion.

Reviewer #3:

This study presents climate zones based on realized niches of the world's terrestrial vertebrates using network methods. In general, I found the analyses to be sound and conclusions valid, although I do have some few reservations. For example, the authors explored the effect of mean annual potential evapotranspiration and annual precipitation as proxies for energy and water inputs, and therefore determinants of tetrapod diversity. The choice of these metrics comes a bit “off-the-shelf” without unpacking the structural reasons for the potential of other variables such as elevation/altitude or latitude driving tetrapod niches, all of which are key determinants for tetrapod diversity. Other aspects that might be worth incorporating or at least discussing is how other regionalizations such as the recent update of Wallace's zoogeographic regions by Holt et al., 2013, fit into the climate niches identified in this study. How can they inform conservation in the real-world? I also provide some suggestions for improvement throughout the manuscript as detailed below.

Abstract:

Please remove the comparison to plants to avoid any controversy on whether or not plants capture the climate of a region than animal taxa or vice versa. The Abstract will read just fine without the last sentence.

Introduction:

The effect of measurement scale on patterns of biodiversity is increasingly becoming a topic of growing interest in ecology and not acknowledging it can lead to spurious conclusions (see some examples of addressing this topic):

Levin, S. A. The problem of pattern and scale in ecology. Ecology 73, 1943-1967 (1992).

Rahbek, C. & Graves, G. R. Multiscale assessment of patterns of avian species richness. Proc. Natl Acad. Sci. USA 98, 4534-4539 (2001).

Jarzyna, M. A. & Jetz, W. Taxonomic and functional diversity change is scale dependent. Nat. Commun. 9, 2565 (2018).

Daru, B.H., Farooq, H., Antonelli, A. & Faurby, S. (2020) Endemism patterns are scale dependent. Nature Communications 11: 2115

It would be important to discuss how the analysis done here can be sensitive to the measurement scale.

Figure 1: This is a nice figure and would help the reader get the message of the paper even without reading the entire manuscript! I like it. However, it appears that the coloring in panel D is switched. It should be the other way around.

Results

Animal vs plant climatic regions

I found the comparison of the Tetrapoda classes in this study to that of Köppen's plant-based regions strange. Köppen's plant-based regions were based on pioneer plant classifications and not necessarily based on the realized niches of the over 380,000 species of plants across the globe, whereas the current study uses carefully vetted updated range maps of over 26,000 vertebrates from the IUCN. I suggest the authors minimize such plant vs animal comparisons.

Figure 3: "Tetrapoda". Since this is the central figure of the paper, the authors should also indicate the names (or suggested names) of the 16 climatic regions directly on the map, perhaps in a separate blown-out figure.

Discussion:

In addition to the caveats already discussed, another caveat worthy of discussion is how sensitive are the climatic regions to the measurement scale along grain size, spatial extent or even taxonomic treatment? See similar comments above.

[Editors' note: further revisions were suggested prior to acceptance, as described below.]

Thank you for resubmitting your work entitled "Regularities in species' niches reveal the world's climate regions" for further consideration by *eLife*.

Your article has been re-evaluated, and although we found it much improved, there are some remaining issues that need to be addressed before acceptance, as outlined below:

Reviewer #2:

I think the responses to reviewers have been very thorough.

Reviewer #3:

The work presented by Calatayud and co-authors on a zoogeographic classification of climate regimes is novel. The climatic determinants shaping animal distributions is largely unknown. Network modelling of realized niche domains of 26,000 vertebrates – accounting for the climatic conditions a species experiences within its range – reveals new classification of global climate regimes.

I enjoyed this second draft, just like I did the first time. The authors addressed some of my comments in the previous version of the manuscript. Thank you. However, a few other comments were not well addressed at least not entirely:

1) The issue of scale: The authors acknowledge incorporating how they have dealt with sensitivity of the analyses to the scale of measure in the Materials and methods. However, not only did I feel like my comments were not well understood, these were also not discussed in the Discussion. How do the findings vary across grain sizes or spatial extents?

For instance, the authors stated that "Extracting the climatic values that a species range covers from a high-resolution climatic raster (such as 0.08°) may reduce commission errors at the species range's borders, but otherwise increases this error. Extracting the climatic values that a species range covers from a high-resolution climatic raster (such as 0.08°) may reduce commission errors at the species range's borders, but otherwise increases this error."

While this assumption may be true to some extent, however, the authors did not test this empirically.

Another example of the need to test for scale effects is how the authors stated that: "To alleviate the effects of these potential errors, we first extracted the climatic values from the high-resolution rasters (0.08°). Then, we computed the average climatic values among selected raster pixels located within cells of 0.5 degrees. In this way, we reduced the effects of commission errors both at the borders of and inside species ranges. We also conducted a bootstrap significance test that takes the uncertainty of species ranges into account (see below)."

In terms of grain size for instance, it is well known that significant climate variation can exist even within a 0.5 degree cell (equivalent to ~50 km) that spans mountain peaks and valleys, and where temperatures can vary over 20°C. In a case where the average of 0.08 degree cells are taken within a 0.5 degree cell, is it unlikely that the average represents the climatic conditions of the grid cell. I imagine this will also be quite different at a grid resolution of 1.0 degree, 2.0 degrees, and/or 5 degrees etc. It is not clear how this was addressed in the Materials and methods and discussed in the Discussion.

In terms of the spatial extent, will the findings still hold up if this was analyzed at a regional continental scale or local country level (and varying along the grain sizes mentioned above)? Adding a test and a discussion along these lines would be important.

2) Climatic transition zones

The authors analyzed an important issue about climate transition zones which can represent places where different climate regimes admix and potentially provide source for novel species interactions, and potentially high diversity. However, the authors miss a great opportunity to return to this aspect later in the Discussion. I was hoping for some real discussion about this where you discussed regions of milder climates. Do the transition zones correspond to these regions of milder climates, and/or what do they mean?

---

## [Author Response]

Reviewer #1:My comments are meant to be constructive, and I hope they will be helpful as you revise your manuscript.Overall opinionIn this manuscript, Calatayud et al. use a network-based approach to identify climatic regions based on tetrapods occurrence. With these in hand, they assess their congruence with the regions of the Köppen-Geiger climate classification (based on climate and plant distributions) and they also assess their specificity. Overall, I think that this study has a lot of merit and I truly value the work done. However, the current version of manuscript requires some work before being publishable. I have several major concerns that I develop below.Lack of clarityThe lack of clarity is twofold. First, some sentences are just not clear and need to be rephrased. Below are three examples that all concern the description of the method:

Thanks for raising this concern. We have rephrased several parts of the methodology section for clarity and precision.

– Subsection “Climatic transition zones” I had to read this several times to understand the meaning of it. A formal definition of D is missing (what are the species included in it).

We have rewritten the section Climatic transition zones for clarity, and included a definition of *D*. The beginning of the section now reads:

“The joint classification of climatic bins and species into domains *D* allowed us to calculate the bins’ specificity. Though species belong to single domains, typically together with the bins to which they have many and strong links, they may also have links to bins in other domains. Bins that contain species from different domains have low specificity and form a transition zone between domains (Calatayud et al., 2019; Bernardo-Madrid et al., 2019)…”

– Subsection “Geographical signal” I think the equations are valid for 1 domain, but this is not made clear (e.g. how what makes index $i$ disappear from (2) to (1))

We have also rewritten the section Geographic signal and revised the equation. We define the raster cell set *Q_i_*as the set of all geographical raster cells *q* with the climate of climatic bin *i* in the end of section Climatic transition zones. The beginning of section Geographical signal now reads:

“To quantify the geographical signal, we compared the geographically projected specificity *S^P^*with the actual specificity based on the pool of species that cooccur geographically. Large differences between the species co-occurring in the climatic and geographic spaces indicate a strong geographic signal. For instance, when species assigned to the same domain co-occur only in a portion of the corresponding geographical space, geographical areas with mismatching species contribute to a strong geographical signal. Areas that host most of the species associated with a niche domain have a higher actual specificity than the projected indicates…”

– The text in Appendix 1 is hard to read.

We improved Appendix 1 following your suggestions. We believe the appendix is clearer now.

Because of this lack of clarity, I am not 100% sure about what has actually been done. It might just be a matter of being careful with the annotations and carefully detailing all the steps, so it might not be much work, but this needs to be done.

Your useful suggestions helped us clarify the methodology.

Second, some details are missing in critical parts. For example:– "Using a hierarchical network clustering algorithm (Rosvall and Bergstrom, 2008, 2011)". This is a critical step and the procedure is published, but I think we need more details to understand what is done, i.e. the rational of the algorithm and why it is suited for this analysis.

Thanks for raising this concern. We added an explanation of the basis of the algorithm and our reasons for choosing it in the Materials and methods section.

“To identify the niche domains, we used the hierarchical version of the community detection algorithm known as Infomap (Rosvall and Bergstrom, 2008, 2011). Infomap capitalizes on the minimum description length principle of information theory, which equates finding regularities and compression: The model that finds most regularities in a given set of data can compress the data the most (Rissanen, 1978). In our case, modules of highly interconnected climatic bins and species form the regularities, and describing the network with an optimal set of communities corresponds to minimizing the description length Bernardo-Madrid et al. (2019); Rosvall and Bergstrom (2008). Among the many community detection algorithms available, we used Infomap because it can find hierarchically nested communities and is known for its high performance (Lancichinetti and Fortunato, 2009), also for regionalizations (Bernardo-Madrid et al., 2019; Bloomfield et al., 2018; Vilhena and Antonelli, 2015).”

Also, I don't understand where the assessment of the domain's robustness is done in the Appendix.

Thanks for pointing out this problem. It was a mistake since we explain the robustness analysis in the Materials and methods section. We changed accordingly.

– AMI is not (yet) frequently used in ecology so I would remind the reader what is it otherwise the reader may not understand the meaning of a AMI of 0.7 (e.g.).

Agree. We included an explanation about AMI:

“AMI measures the mutual information between two partitions, the classifications of raster pixels into climatic regions in this case, correcting for the similarity between partitions that are just due to chance (Vinh et al., 2010). The index is 1 when the partitions are equal and tends to 0 otherwise.”

My point being that the right level of methodological is not always provided. A few more sentences may do the jobs.Why this is not all trivial?Full disclaimer, I don't think the results are trivial. But at first sight, one may think so.After my first read of the manuscript, I thought that the two following assertions of the Introduction were contradictory– "abiotic conditions determine species ranges"– "However, the fact that plant species are good indicators of general climatic conditions does not necessarily imply that such conditions shape the distribution of other organisms in the same manner"If "abiotic conditions determine species ranges" and if the reference classification is based on plant distributions, given the variety of ecological links between plants and tetrapodes, then the distribution of tetrapodes communities (assuming there are properly defined) and the reference distribution (Köppen) should be very similar. And actually they are similar as it is showed in Figure 3. But after thinking more about this I convinced myself otherwise, partly thanks to the manuscript, but not only. And this is the issue: the authors should better explain why this is not all trivial in the Introduction, they should explain why one could expect significant discrepancies between the two distributions.

In the Introduction, we clarified that while abiotic conditions can determine species ranges, it is expectable that climatic determinants of plant and animal ranges differ due to the large physiological differences between these groups. Moreover, we supported this statement with previous evidence on different climatic correlates of diversity patterns of plants and animals. We hope the reviewer find this justification more compelling.

“…the fact that plant species are good indicators of general climatic conditions does not necessarily imply that such conditions shape the distribution of other organisms in the same manner. For instance, given the large physiological differences between plants and animals, it is expectable that the climatic determinants of their ranges will also be different. Indeed, while diversity patterns of plant and animals are to some extent congruent, the climatic correlates of these patterns vary (Whittaker et al., 2007; Qian and Ricklefs, 2008). Hence, if climatic adaptations vary among taxa, the boundaries defining their climate types will also vary.”

Impact statementImportantly enough, I don't think that the impact statement : "Similarities in climatic niches of terrestrial vertebrates indicate the Earth's climate regions, which substantially differ from previous plant-based climate classifications." is a fair depiction of the results and it is actually contradictory with, inter alia, the results presented in Figure 3 (the caption of which reads "Tetrapoda groups and Köppen's climatic regions are largely congruent"). Unless I have miss something.

We are thankful for this comment. The reviewer is right; these two statements are contradictory. We removed the adjective phrases “substantially” and “largely congruent” so that the readers can form their own opinions.

DiscussionI must say that the Discussion section includes various bold statements that the authors, for instance:"Our results bring us closer to a definition of climatic regions that represent active factors for the organization and evolution of life."Quite frankly, after reading the manuscript, I am not entirely sure why. I would rather discourage the authors from making such statements.

We agree that the quoted sentence was too bold. We hedged the statement:

“Our results and data-driven methodology have the potential to bring us closer to a definition of climatic regions that represent active factors for the spatial organization of life.”

We carefully searched for other sentences that may also be controversial and removed or rephrased them.

Reviewer #2:This is a brilliant piece of work, filling in a massive gap in biogeography. I think for a first shot, including just one measure of energy and one for water is perfectly fine. The writing is good and the logic mostly consistent, with very slight lapses in the discussion.Reviewer #3:This study presents climate zones based on realized niches of the world's terrestrial vertebrates using network methods. In general, I found the analyses to be sound and conclusions valid, although I do have some few reservations. For example, the authors explored the effect of mean annual potential evapotranspiration and annual precipitation as proxies for energy and water inputs, and therefore determinants of tetrapod diversity. The choice of these metrics comes a bit “off-the-shelf” without unpacking the structural reasons for the potential of other variables such as elevation/altitude or latitude driving tetrapod niches, all of which are key determinants for tetrapod diversity.

It is certainly true that other variables may turn important to define tetrapods’ niches. Indeed, we explicitly discussed this possibility at various points in the Discussion section:

“…alternative variables, such as seasonal changes in energy and precipitation (Köppen and Geiger, 1930) may also influence species inhabiting temperate regions. Including these variables might help to further separate temperate species pools across the climatic space.”

“…but other climatic variables might refine some of the least supported regions.”

Nevertheless, and in line with reviewer 2’s comments, annual precipitation and mean annual potential evapotranspiration are sufficient variables to capture water and energy aspects of species niches. These aspects are arguably the most important to understand the distribution of species in general (Hawkins et al., 2003; Currie, 1991; Tingley et al., 2009; Gouveia et al., 2014; Pie et al., 2017; Cooper et al., 2011) and, therefore, precipitation and PET are among the best variables to identify global climate regions (Thornthwaite, 1948), which is our final goal.

Finally, we believe there are reasons for not considering elevation and latitude as plausible variables in our analysis. Elevation and latitude are indeed important correlates of tetrapods diversity. Yet, it is widely accepted that they can only be correlates, but not drivers, of such diversity because these variables *per se* do not have direct biological meaning in most instances. Latitude and elevation are biologically indirect topo/geographical variables that can inform about variations in biologically direct environmental ones such as PET, precipitation, temperature, atmospheric pressure, and so on. The latter directly shape species niches and derived diversity patterns. Thus, while latitude and elevation may work as surrogates of climatic variables, they are not climatic variables (Körner, 2007).

Since we aim to describe climate regions, we think direct climatic variables are superior to latitude and elevation. Indeed, to our knowledge, there is no any previous study where these variables have been used or considered when delimiting climate regions and, thus, we are not totally convinced on the need of explaining their potential for our purposes.

We hope you find this arguments convincing. Otherwise we are open to further suggestions to improve our manuscript.

Other aspects that might be worth incorporating or at least discussing is how other regionalizations such as the recent update of Wallace's zoogeographic regions by Holt et al., 2013, fit into the climate niches identified in this study.

Thanks for this comment, which helped us to improve the discussion. Combining information from previous regionalizations based on species or lineages pools with the climate regions we proposed can definitively be a fruitful exercise. We added some sentences on the potential of such a combination in the Discussion section:

“Moreover, combining niche domains with bioregions based on pools of species (Bernardo-Madrid et al., 2019) or lineages (Holt et al., 2013) can also help answer several relevant questions: Are introductions more common between bioregions with the same climate types? To what extent does the current climate govern bioregions? Do ecological and evolutionary processes differ between different bioregions with same climate types (Moncrieff et al., 2015)?”

We did not directly compare these regionalizations with the climate regions, as this would require a battery of analyses and interpretations far beyond this study’s scope.

How can they inform conservation in the real-world?

This is a good question. To be more conservative regarding using the climatic regions for conservation planning in the real world, we rephrased the paragraph related to it. We changed “conservation plannings” to “conservation studies” since there are direct applications of the climatic regions for such studies. We also reformulated the question related to conservation that using the climatic regions can be answered and cited a relevant study on the topic.

“…the niche domains and their associated species pools and climatic regions can be used as a basis for ecological, evolutionary and conservation studies concerning Tetrapoda. …Are the species belonging to different niche domains similarly conserved or protected or both (Hanson et al., 2020)? …”

I also provide some suggestions for improvement throughout the manuscript as detailed below.Abstract:Please remove the comparison to plants to avoid any controversy on whether or not plants capture the climate of a region than animal taxa or vice versa. The Abstract will read just fine without the last sentence.

We rephrased this sentence to avoid the apparent controversy:

“…call for caution when using general climate classifications to study the ecology, evolution or conservation of specific taxa.”

Introduction:The effect of measurement scale on patterns of biodiversity is increasingly becoming a topic of growing interest in ecology and not acknowledging it can lead to spurious conclusions (see some examples of addressing this topic):Levin, S. A. The problem of pattern and scale in ecology. Ecology 73, 1943-1967 (1992).Rahbek, C. & Graves, G. R. Multiscale assessment of patterns of avian species richness. Proc. Natl Acad. Sci. USA 98, 4534-4539 (2001).Jarzyna, M. A. & Jetz, W. Taxonomic and functional diversity change is scale dependent. Nat. Commun. 9, 2565 (2018).Daru, B.H., Farooq, H., Antonelli, A. & Faurby, S. (2020) Endemism patterns are scale dependent. Nature Communications 11: 2115It would be important to discuss how the analysis done here can be sensitive to the measurement scale.

This a critical comment. Indeed, we were very cautious when dealing with potential scales issues at all relevant points of our analyses.

First, we explained the potential commission issues related to the grain size used to characterize the species climatic niches:

“… Extracting the climatic values that a species range covers from a high resolution climatic raster (such as 0.08°) may reduce commission errors at the species range’s borders but otherwise increases this error. Conversely, extracting climatic values from a coarser raster can reduce the influence of commission errors over the areas inside a range but increase them over the borders.”

Moreover, we applied an elaborated procedure (including a bootstrap sensibility analysis) to deal with this issue:

“…we first extracted the climatic values from the high-resolution rasters (0.08°). Then, we computed the average climatic values among selected raster pixels located within cells of 0.5 degrees. In this way, we reduced the effects of commission errors both at the borders of and inside species ranges. We also conducted a bootstrap significance test that takes the uncertainty of species ranges into account.”

Second, we explained the potential issues related to the grain size used to divide the climatic space:

“…dividing the space into too few intervals destroys information, whereas using too many divisions can generate niche domains with only a few species.”

To solve this problem, we developed the optimization procedure explained in Appendix 1.

Third, differences due to taxonomic scale are at the core of the study because we compared the climatic regions of Tetrapoda class with the ones of its subclasses.

Finally, regarding the spatial and climatic extent, we are constrained from above by the study’s global perspective. On the other hand, the potential effects of reducing the extent, which would imply focusing on particular geographic regions, are also at the core of the study because we considered the possible discrepancies between regions when analyzing the geographical signal.

For the reasons above, we think the scale issues related to our analyses have been treated and discussed adequately and extensively.

To further acknowledge the critical role scale plays for analyzing ecological patterns, and why we care so much about it in the analysis, we have added the suggested references when we begin dealing with scale in the section Niche characterization in Materials and methods.

Figure 1: This is a nice figure and would help the reader get the message of the paper even without reading the entire manuscript! I like it. However, it appears that the coloring in panel D is switched. It should be the other way around.

This is right. Thank you very much for noticing this error and the positive comments. We fixed the figure.

ResultsAnimal vs plant climatic regionsI found the comparison of the Tetrapoda classes in this study to that of Köppen's plant-based regions strange. Köppen's plant-based regions were based on pioneer plant classifications and not necessarily based on the realized niches of the over 380,000 species of plants across the globe, whereas the current study uses carefully vetted updated range maps of over 26,000 vertebrates from the IUCN. I suggest the authors minimize such plant vs animal comparisons.

Indeed, Köppen based his climatic regions on the distribution of vegetation types rather than on species. Still, we think this comparison is valuable because Köppen’s system is nowadays the reference climate classification. It is also the most widely used, even in studies dealing with animals. We think it would be more strange to present a new classification without comparing the previously established one. To include this information and motivate this comparison, we wrote at the beginning of the section at question:

“While Köppen’s climate classification is based on expert-knowledge on vegetation physiognomy and the distribution of vegetation types, its wide use makes it worthwhile to compare the climate regions derived from this classification system to the ones produced here.”

Moreover, to minimize the animal vs. plant comparison, we renamed the section to “Animal versus previous plant-based climatic regions”

Figure 3: "Tetrapoda". Since this is the central figure of the paper, the authors should also indicate the names (or suggested names) of the 16 climatic regions directly on the map, perhaps in a separate blown-out figure.

Agree. Thanks for this comment. We added the names of the regions in Figure 3B.

Discussion:In addition to the caveats already discussed, another caveat worthy of discussion is how sensitive are the climatic regions to the measurement scale along grain size, spatial extent or even taxonomic treatment? See similar comments above.

As previously mentioned, we discussed and treated the sensitivity to different scale choices in other relevant places in the manuscript. We think we have adequately and extensively considered scale issues as detailed above.

[Editors' note: further revisions were suggested prior to acceptance, as described below.]

Reviewer #3:The work presented by Calatayud and co-authors on a zoogeographic classification of climate regimes is novel. The climatic determinants shaping animal distributions is largely unknown. Network modelling of realized niche domains of 26,000 vertebrates – accounting for the climatic conditions a species experiences within its range – reveals new classification of global climate regimes.I enjoyed this second draft, just like I did the first time. The authors addressed some of my comments in the previous version of the manuscript. Thank you. However, a few other comments were not well addressed at least not entirely:1) The issue of scale: The authors acknowledge incorporating how they have dealt with sensitivity of the analyses to the scale of measure in the Materials and methods. However, not only did I feel like my comments were not well understood, these were also not discussed in the Discussion. How do the findings vary across grain sizes or spatial extents?

Thanks for raising this issue again and for the detailed explanation. We must admit that we were surprised by your concerns about scale issues during the first revision round because thorough sensitivity analysis is at the core of our work. For example, our statistical assessments go beyond those in recent studies on related topics and data (e.g. Daru et al., 2017; Holt et al., 2013; Ficetola et al., 2017; Gardner et al., 2020). With your expanded explanation and detailed examples, we realized that we had been victims of the curse of knowledge with unclear descriptions on how we dealt with scale issues. As explained below, we clarified these points throughout the text and included several lines about scale issues in the Discussion.

For instance, the authors stated that "Extracting the climatic values that a species range covers from a high-resolution climatic raster (such as 0.08°) may reduce commission errors at the species range's borders, but otherwise increases this error. Extracting the climatic values that a species range covers from a high-resolution climatic raster (such as 0.08°) may reduce commission errors at the species range's borders, but otherwise increases this error."While this assumption may be true to some extent, however, the authors did not test this empirically.

Correct, we have not explicitly and empirically tested this argument because it aimed to alleviate uncertainties related to range maps based on most species’ theoretically expected situation. We have rewritten the paragraph to clarify this point, adding some citations to studies with similar reasoning:

“Because range maps typically represent the species’ maximum geographical extent (Rondinini et al., 2006; La Sorte and Hawkins, 2007), extracting the climatic values that a species range covers from a well-fitting 0.08° high-resolution climatic raster can reduce commission errors at the species range’s borders. But with many pixels inside the species range, the noise from extreme and unrepresentative climatic values can increase the error (La Sorte and Hawkins, 2007). Conversely, extracting climatic values from a coarser raster can reduce the commission errors inside a range by averaging out extreme values at the cost of increasing commission errors over the borders.”

Finally, we now mention this uncertainty issue in the Discussion:

“…our domains represent portions of the realized climatic niche space, which entails two sources of uncertainty. On the one hand, the estimation of realized niches depends on the data quality, scale, and treatment choices. On the other hand, the realized niche space may be influenced by historical, geographical, and biotic factors beyond pure climate (Warren et al., 2014; Calatayud et al., 2019b; Soberón, 2007). Hence, using fundamental rather than realized niches may also improve the accuracy of defining climatic regions.”

Please notice that most of the apparent issues of scale effects we are discussing are related to the estimation of realized niches. Hence, using the fundamental rather than the realized niche would be the best way to overcome them.

Another example of the need to test for scale effects is how the authors stated that: "To alleviate the effects of these potential errors, we first extracted the climatic values from the high-resolution rasters (0.08°). Then, we computed the average climatic values among selected raster pixels located within cells of 0.5 degrees. In this way, we reduced the effects of commission errors both at the borders of and inside species ranges. We also conducted a bootstrap significance test that takes the uncertainty of species ranges into account (see below)."In terms of grain size for instance, it is well known that significant climate variation can exist even within a 0.5 degree cell (equivalent to ~50 km) that spans mountain peaks and valleys, and where temperatures can vary over 20°C. In a case where the average of 0.08 degree cells are taken within a 0.5 degree cell, is it unlikely that the average represents the climatic conditions of the grid cell. I imagine this will also be quite different at a grid resolution of 1.0 degree, 2.0 degrees, and/or 5 degrees etc. It is not clear how this was addressed in the Materials and methods and discussed in the Discussion.

We agree that this is a potential issue in the many biogeographical and macroecological studies using grid cells with the mentioned grain sizes. While averaged climatic values might not represent well the climates of such large cells, it is generally accepted that the effects of this misrepresentation mostly vanish at large scales. For example, cells in the tropics are on average warmer than cells close to the poles such that the climatic variability between cells at large scales is high enough to override the effects of the within-cell variability. But even when taking this difference into account, we agree that it is crucial to test the impact of distorted climatic means in coarse-grained grids. Using different grid resolutions and comparing the results is one way to test the robustness. To measure the effects of not only within-cell climatic variability but also non-normal distributions, instead, we used non-parametric bootstrap resampling: We resampled climatic values within cells with replacement, reclassified the data, and estimated the probability of obtaining similar climatic regions. Overall, we found high bootstrap probabilities showing that the effects of distorted climatic means from coarse-grained grids are small in our case.

We now explain this sensitivity analysis better in the Materials and methods section:

“… mean climatic values may not accurately represent the cells’ climates when there is high climate variability, or the values are non-normally distributed. Comparing results obtained from different cell sizes is an indirect way to assess the influence of distorted mean values. Instead, we chose to directly explore the effects of high climatic variability and non-normal distributions with nonparametric bootstrap analysis: We resampled climatic values within 0.5^◦^ cells with replacement (see below).”

“… we conducted a bootstrap analysis. For each species, we resampled with replacement from the distribution of climatic values within species ranges at a resolution of 0.08°. We averaged climatic values occurring within 0.5^◦^ cells and calculated the proportion of occurrences in each climatic bin. With resampled data from all species, we generated a bootstrapped network and clustered it with Infomap 1000 times. We repeated this procedure for 100 bootstrap networks and followed the community-stability approach proposed in Calatayud et al., 2019a, to calculate the niche domains’ support.”

Finally, since this issue is also related to the estimation of realized niches, we now mention them together in the Discussion.

In terms of the spatial extent, will the findings still hold up if this was analyzed at a regional continental scale or local country level (and varying along the grain sizes mentioned above)? Adding a test and a discussion along these lines would be important.

We agree that delineating climate regions at smaller extents can be of interest. However, such an analysis is beyond the scope of our manuscript’s global perspective. As explained above, the bootstrap analysis more directly assesses the issues that would motivate the analyses at different grain sizes. Taken together, and following the first revision round’s suggestions, we have mentioned this idea in the Discussion rather than adding more tests.

“…the geographical signal idea could also help unravel idiosyncratic processes occurring at smaller scales by comparing local climate regions with global or regional ones.”

2) Climatic transition zonesThe authors analyzed an important issue about climate transition zones which can represent places where different climate regimes admix and potentially provide source for novel species interactions, and potentially high diversity. However, the authors miss a great opportunity to return to this aspect later in the Discussion. I was hoping for some real discussion about this where you discussed regions of milder climates. Do the transition zones correspond to these regions of milder climates, and/or what do they mean?

We appreciate this new comment as it allows us to clarify the meaning of the transition zones. We have reworded the text to explain the transition zones’ importance in our study. Specifically, we make the connection between “specificity” and “transitivity” more transparent. We believe the Results and Discussion now provide clear answers to your questions:

“Climatic conditions corresponding to diffuse transitions should present low specificity levels to the domain where they belong (Figure 1). Hence, we can consider specificity as the opposite of transitivity… As expected, lower specificity values were in general associated with the boundaries of the climatic regions (Figure 4A and Appendix 1—figure 2). Beyond boundaries, our results also revealed that harsh conditions, such as desert and continental-polar climates (E^H^W^L^ and E^L^W^L^), present the highest specificity levels, regardless of the group (Figure 4A and Appendix 1—figure 2), reflecting the difficulty of colonizing these climates. Conversely, temperate regions showed the lowest levels of specificity.”

In the Discussion, we now write:

“Conversely, milder climatic conditions, especially temperate climates, showed the lowest statistical support and congruence across groups, as well as the highest transitivity. These results indicate that temperate climates are more difficult to classify (low support and congruence) due to the overlap in the climatic space of species pools with different climatic optima (high transitivity).”

Finally, we mention that:

“…the exact causes and consequences of the climatic transition zones call for

future investigation.”

We prefer to be conservative and avoid a detailed but speculative discussion about the causes and consequences of transition zones. Such claims would require an in-depth empirical evaluation of different causal factors beyond the scope of this study.